# Coupled hydrological and hydrodynamic modelling application for climate change impact assessment in the Nemunas River watershed–Curonian Lagoon–south-eastern Baltic Sea continuum

Rasa Idzelytė [1], Natalja Čerkasova [1,2], Jovita Mėžinė [1], Toma Dabulevičienė [1], Artūras
Razinkovas-Baziukas [1], Ali Ertürk [1,3], Georg Umgiesser [1,4]

[1]Marine Research Institute, Klaipėda University, Klaipėda, 92294, Lithuania
[2]Texas A&M AgriLife Research, Blackland Research and Extension Center, Temple, TX 76502, USA
[3]Department of Inland Water Resources and Management, Istanbul University, Istanbul, 34134, Turkey
[4]CNR–National Research Council of Italy, ISMAR–Institute of Marine Sciences, Venice, 30122, Italy

*Correspondence to*: Georg Umgiesser (georg.umgiesser@ismar.cnr.it)

**Abstract.** We analyse the cumulative impacts of climate change in a complex basin-lagoon-sea system continuum, which covers the Nemunas River basin, Curonian Lagoon, and the south-eastern part of the Baltic Sea. A unique state-of-the-art coupled modelling system was developed using hydrological and hydrodynamic models. Results of four regional downscaled models from the Rossby Centre high-resolution regional atmospheric climate model have been bias-corrected using in situ measurements, and were used as forcing to assess the changes that the continuum will undergo until the end of this century. Results show that the Curonian Lagoon will be subjected to higher river discharges that in turn increase the outgoing fluxes into the Baltic Sea. Through these higher fluxes, both the water residence time and saltwater intrusion into the lagoon event frequency will decrease. Most of these changes will be more pronounced in the northern part of the lagoon, which is more likely to be influenced by the variations in the Nemunas River discharge. Its delta area may be susceptible to flooding as a result of the elevated discharge during winter. The southern part of the lagoon will experience lesser changes. While water temperatures in the entire lagoon and the south-eastern Baltic Sea will steadily increase, and salinity will decrease, the foreseen changes in physical characteristics will not cause significant shifts in the ecosystem functioning, but may affect the nutrient retention capacity. However, some ecosystem services such as ice fishing are expected to vanish completely due to the loss of ice cover.

## 1 Introduction

Climate change and increasing anthropogenic pressures are expected to cause drastic changes in the global environment in the near future, and thus may affect the discharge and hydrological regime of rivers (Middelkoop et al., 2001), lagoons (Jakimavičius and Kriaučiūnienė, 2020; Anthony et al., 2009) and regional seas (Holt et al., 2016). Coastal lagoons and estuaries, forming a continuum between continental and marine aquatic ecosystems (Pérez-Ruzafa et al., 2011) play an important ecological and socio-economic role (Newton et al., 2018; Camacho-Ibar, and Rivera-Monroy, 2014). They support

important habitats and rich biodiversity (Newton et al., 2018), are nursery grounds and nutrient sources for nearshore fish communities (Meynecke et al., 2007), and are important areas for various biogeochemical processes (Watson et al., 2020; Maher et al., 2019). Coastal lagoons also provide a set of ecosystem services from the physical use of land-/seascape to cultural or aesthetic value (Inácio et al., 2018).

In general, lagoon systems comprise 5.3% of the European coastlines with the south-eastern coast of the Baltic being one of the major lagoon regions in Europe (de Wit, 2012). Due to their geographic location, lagoons become very vulnerable to the climate change-related key drivers and arising therefrom physical, ecological, and associated societal disturbances, such as changes in flushing regime, freshwater inputs, water chemistry or loss in communities (Anthony et al., 2009).

The Baltic Sea, together with its largest lagoon, the Curonian Lagoon, and the Nemunas River watershed are at the forefront

of the climate change impact, which is already evident here through the changes in ice conditions in the sea (Merkouriadi and Leppäranta, 2014) and the lagoon (Idzelytė et al., 2019), increase in sea surface temperature (Kniebusch et al., 2019; Belkin, 2009) or redistribution of river runoff over the year (Meier et al., 2022a).

Additional climate change impacts might manifest themselves slowly over decades (Lillebø, 2015) making the management of freshwater and transitional waters a challenge for policymakers, decision makers, and scientists. For example, climate

change-induced increase in precipitation will result in an increased nutrient leaching from agricultural soils, forests, and other land uses (Pihlainen et al., 2020) making it even more difficult to combat eutrophication – a prevalent and serious environmental problem in the Baltic Sea, affecting both the coastal waters and the open sea (McCrackin et al., 2018; Vigouroux et al., 2012). Hence, modelling becomes increasingly important for understanding and projecting climate change impacts, also for supporting the formulation of management measures (Anthony et al., 2009; Vohland et al., 2014).

The first climate change scenario simulations for the Baltic Sea were carried out more than 20 years ago (Meier, 2002a, b; Omstedt et al., 2000). ln 2008, the first assessment of the climate change in the Baltic Sea region was prepared with the main conclusions that by the end of 2100 the annual mean sea surface temperature will be 2–4 °C higher while the ice cover and salinity values will decrease (BACC Author Team, 2008). One of the new key findings of the 2nd climate change assessment (BACC II Author Team, 2015) was that due to the large bias in the water balance projections the projected changes of the

Baltic Sea salinity remained uncertain, i.e., it is unclear if the Baltic Sea becomes more or less saline (von Storch et al., 2015). Nevertheless, climate change is consequently going to affect marine ecosystem and may reduce its resilience.

The magnitude and severity of climate change impacts will vary considerably within the Baltic Sea region (Graham, 2004) making basin-specific studies of high importance for understanding climate change induced impacts on a local scale. Likewise, with increasing computational capacity, the multidisciplinary modelling studies were progressively evolving during the past

few decades (Rodrigues et al., 2015). Thus, the need for a combination of different aquatic processes became essential.

The coupling of different numerical models allows researchers to replicate the environmental processes for better assessment of the physical parameters' response to changing climate. Regarding the climate applications, the inclusion of different models, e.g. hydrological and hydrodynamic, are particularly important since it provides necessary feedback between the atmosphere land, and ocean, reducing the biases of such estimations. Notably, coupling of a hydrological model into the large modelling

systems provides connectivity between the hydrology of the land surface and ocean or sea, making it a crucial component of the coupled system (Hagemann et al., 2020). Choice of climate scenarios is an important consideration in climate research as well. The Representative Concentration Pathway (RCP) 4.5 and 8.5 scenarios represent two different pathways of future greenhouse gas emissions and are commonly used in climate modelling (Thomson et al., 2011; Riahi et al., 2011). Comparing these scenarios allows researchers to assess the potential effects of moderate climate mitigation efforts. It is important to note 70 that neither scenario is necessarily more "realistic" than the other, as both represent possible pathways of future emissions depending on societal choices and actions.

In this article, we present a framework of coupled hydrological and hydrodynamic models that are able to simulate the contribution of the entire Nemunas River watershed to the Curonian Lagoon. Moreover, the Curonian Lagoon and the south-eastern Baltic Sea are covered by a finite element grid that resolves its hydrodynamic and physical parameters. A seamless 75 integration of two models represents the watershed–lagoon–sea continuum for the simulations of the present state and future scenarios based on the ensemble of meteorological datasets produced by climate models.

## 2 Materials and methods

### 2.1 Study area

The study site covers three domains – part of the south-eastern Baltic Sea region together with the Curonian Lagoon and the 80 Nemunas River watershed (Fig. 1). The drainage basin is shared by five countries: Belarus (48%), Lithuania (46%) and the other 6% is shared by Kaliningrad region of Russian Federation, Poland and Latvia (Gailiušis et al., 2001). The river enters the Curonian Lagoon at its central-eastern part and divides the system into two sub-basins; its southern part belongs to the Russian Federation (Kaliningrad region, 76%) and its northern part to Lithuania (24%).

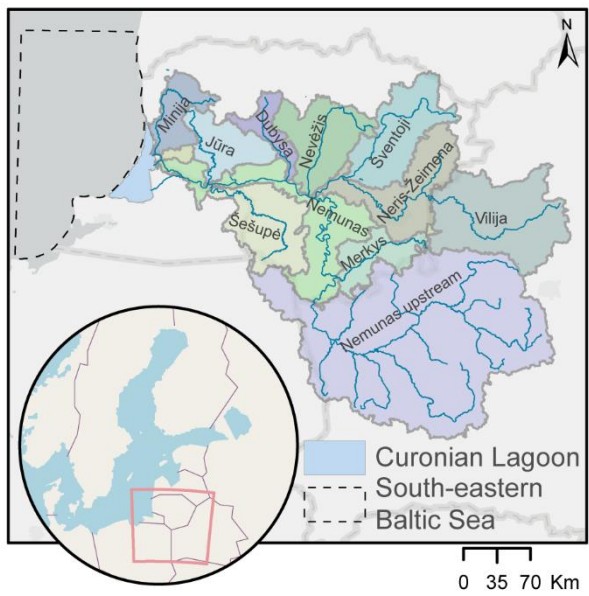

**Figure 1. The Nemunas River watershed (coloured polygons in the mainland are major sub-basins), Curonian Lagoon, and south-eastern Baltic Sea with respect to the entire sea area. Basemap source: ESRI.**

The Nemunas River is the fourth largest river draining into the Baltic Sea and the largest tributary entering the Curonian Lagoon, accounting for ~96% of the total freshwater input into it (Jakimavičius and Kovalenkovienė, 2010). The annual average discharge is 16.4 km$^3$ yr$^{-1}$ (518.3 m$^3$ s$^{-1}$), calculated for the period 2012–2016 (Vybernaite-Lubiene et al., 2018), and

the long-term average of 21.847 km$^3$ yr$^{-1}$ (692 m$^3$ s$^{-1}$) for the period 1960–2007 (Jakimavičius and Kovalenkovienė, 2010).

The Curonian Lagoon is a shallow water body with an average depth of 3.8 m (Gasiūnaitė et al., 2008) located on the south-eastern coast of the Baltic Sea. With a total area of ~1584 km$^2$ (Žaromskis, 1996), it is considered the largest lagoon in Europe. It is a transnational lagoon whose northern part is influenced by the Nemunas River flow and brackish water intrusions from the Baltic Sea, while the southern part is more stagnant, where hydrodynamics are influenced by the wind (Vybernaite-Lubiene

et al., 2022; Ferrarin et al., 2008). The intrusions of the sea water through the narrow (0.4–1.1 km) Klaipėda Strait occur in the northern part of the lagoon and reach up to 20 km southward from the strait (Zemlys et al., 2013).

The coastal area of the south-eastern Baltic Sea is relatively shallow, up to 20 m depth. The lowest water temperatures here are observed in winter months (in Jan and Feb the average water temperature is ~2 °C), while in summer months, e.g., Jul and Aug, the average water temperature is ~18–19 °C (Kozlov et al., 2014). The salinity in the coastal zone is varying between 6–

7 g kg$^{-1}$, although in the plume affected waters from the Curonian Lagoon it can drop drastically to nearly fresh water (0–3 g kg$^{-1}$) (Olenin and Daunys, 2004).

## 2.2 Data

### 2.2.1 Climate projection data

To feed our developed modelling system we used meteorological forcing data acquired from CORDEX (Coordinated Regional Downscaling Experiment) scenarios for Europe from the Rossby Centre high-resolution regional atmospheric climate model (RCA4), which consisted of four sets of simulations (downscaling) driven by four global climate models (specified in Table 1). The datasets consist of cloud cover, solar radiation, precipitation, surface air pressure, relative humidity, air temperature, and wind speed and direction, and covers a historical period of 1970–2005 and projection period of 2006–2100. The projections are derived according to two Representative Concentration Pathway (RCP) scenarios: RCP4.5 and RCP8.5 of the Coupled Model Intercomparison Project - Phase 5 (CMIP5).

| Abbreviation | Model | Institution |
|---|---|---|
| ICHEC | EC-Earth | Irish Centre for High-End Computing |
| IPSL | IPSL-CM 5A-MR | The Institut Pierre-Simon Laplace |
| MOHC | HadGEM2-ES | Met Office Hadley Centre |
| MPI | MPI-ESM-LR | Max Planck Institute for Meteorology |

**Table 1. Summary of the global climate model data sources.**

The applied modelling system uses standard Gregorian calendar, however, MOHC uses 360-day calendar and IPSL 365-day calendar. Therefore, correction was applied to harmonize the input data. MOHC calendar was adjusted by deleting 30 Feb and 29 Feb (if not leap year), and interpolating the 31st of respective months. IPSL was corrected by interpolating the 29 Feb during the leap years.

The bias correction of air temperature and precipitation datasets were done separately for the hydrological and hydrodynamic models by applying the climate data bias correction tool (Gupta et al., 2019), which uses the quantile mapping approach by fitting the daily values to normal distribution function for the air temperature and gamma distribution function for the precipitation. Due to the availability of measurement data, provided by Lithuanian Hydrometeorological Service, the correction period was from 1993 to 2005.

The data from 18 meteorological stations, which are scattered throughout the Republic of Lithuania, were used for the bias correction of the climate data and used as inputs for the hydrological model. For the Belarus region the meteorological grid was constructed according to the methodology presented in the "Renewal of a River Basin Districts Management Plans and Programmes of Measures" project report (PAIC, 2015) and were split in three groups corresponding to the closest meteorological stations in Lithuania of which measured data were used for bias correction. The bias correction of climate input data for the hydrodynamic model was performed from the averaged data of three meteorological stations closest to the northern part of the lagoon that describe the coastal climate conditions.

### 2.2.2 Boundary data

Water level, temperature, and salinity data for the sea boundary of the hydrodynamic model were acquired from a high
resolution regional coupled ocean–sea ice–atmosphere model RCA4–NEMO (Dieterich et al., 2019; Gröger et al., 2019; Wang et al., 2015) developed by the Rossby Centre and the Oceanographic research group at Swedish Meteorological and Hydrological Institute (SMHI). This model was run by the same before-mentioned global climate projections (Table 1).

The bias correction for the data of RCA4-NEMO model was done by using Copernicus Marine Environment Monitoring Service (CMEMS) Baltic Sea Physical Reanalysis product data for the period of 1993–2005. The correction was done by
simply adding the difference between the average values of CMEMS, $C_O$, and RCA4–NEMO data, $C_m$, (Lenderink et al., 2007):

$$C_{BC}(t) = C_m(t) + (C_O - C_m) , \qquad\qquad (1)$$

Bathymetry data of a high-resolution spherical grid topography of the Baltic Sea were used as a bottom boundary (Seifert et al., 2001). Data of ice thickness from ice thermodynamic model ESIM2, run using the meteorological data specified in Table
1, were used as a top boundary during the ice cover season (Idzelytė and Umgiesser, 2021; Tedesco et al., 2009).

### 2.2.3 Watershed-scale data

For the development of the hydrological model many basin-scale datasets are required, i.e., the digital elevation model (DEM), land use and management data, hydrologic grid, soil maps, etc. We obtained the data from several governmental sources in different countries as well as public open access databases. For a full list of datasets and their sources, we refer the reader to
Čerkasova et al. (2021), where the acquired datasets are described in depth. Observed discharge data with varying time step (daily, weekly, by-weekly, and monthly) for the majority of the Nemunas River tributaries as well as the main branch were obtained from the Lithuanian Hydrometeorological Service. Observed nutrient (TN and TP) concentration values for the overlapping periods were provided by the Lithuanian Environmental Protection Agency. The same meteorological forcing data, described in Table 1, was used for the hydrological model, as well as for the hydrodynamic.
Large parts of the watershed are outside of Lithuania, hence open-access data for those regions were identified and used (Čerkasova et al., 2021). Some data had to be manually digitized, i.e., the stream network of the Nemunas watershed outside of the territory of Lithuania. Unfortunately, due to political reasons, we could not acquire any observational data from the Republic of Belarus and the Kaliningrad Region. We resolved the issue by calibrating the model against observed data at the nearest border locations within the territory of Lithuania, provided by abovementioned governmental institutions.

## 2.3 Models

### 2.3.1 Hydrological model

High-resolution basin-scale model has been developed for the entire Nemunas watershed (Fig. 1) and implemented using Soil and Water Assessment Tool (SWAT) using a set of custom tools and scripts. SWAT is a hydrological model that simulates the water balance and quality of watersheds at various spatial scales (Neitsch et al., 2011). It is designed to help land managers and decision-makers evaluate the impact of land use, climate change, and other management practices on water resources. SWAT is widely used around the world for a variety of applications, including water management, erosion control, flood prediction, and water quality management; it has been applied in a range of settings, from small rural watersheds to large river basins (Gassman et al., 2007). The model can simulate a variety of hydrological processes, including surface runoff, infiltration, evapotranspiration, as well as water quality parameters like sediments and nutrients.

SWAT sub-divides the modelled area into sub-regions, called sub-basins, which are further subdivided into Hydrological Response Units (HRU). HRUs are the basic spatial units used in SWAT to represent a homogeneous area of land with respect to soil type, land use, slope, and administrative unit. Each HRU is characterized by a unique combination of these factors, which influence the hydrological and water quality processes within that unit. Moreover, each HRU has a unique set of hydrologic response parameters that govern the water balance within that unit. This approach allows for the simulation of spatially distributed hydrologic processes and water quality, making the model more effective in predicting watershed responses to different land use and management scenarios.

To set up a comprehensive model, we incorporated the topographic information of the area (digital elevation model), land use, land cover data, soil properties, information on water bodies, land management, water use, crop growth, livestock production, administrative units, etc. The model is sub-divided into 11 sub-models, each representing one of the tributaries of the Nemunas River, connected to the main branch (Fig. 2). In total, the model consists of 9012 sub-basins, and 148 212 HRUs. These sub-models are linked from upstream to downstream, where the outputs of two downstream models are directly used as river boundaries for the hydrodynamic model. The hydrologic model was calibrated and validated, and proved to reliably represent the water balance components, such as discharge at major gage stations, as well as nutrient (TP and TN) and sediment load estimates (Čerkasova et al., 2021, 2019, 2018) within the major tributaries of the basin and the main outlet.

Discharge and temperature outputs were extracted for this study; however, water quality variables can be used for further assessment of the water quality in the area. The reader is referred to the study by Čerkasova et al. (2021) detailing the watershed-scale model for the Nemunas river basin and its predictive capabilities.

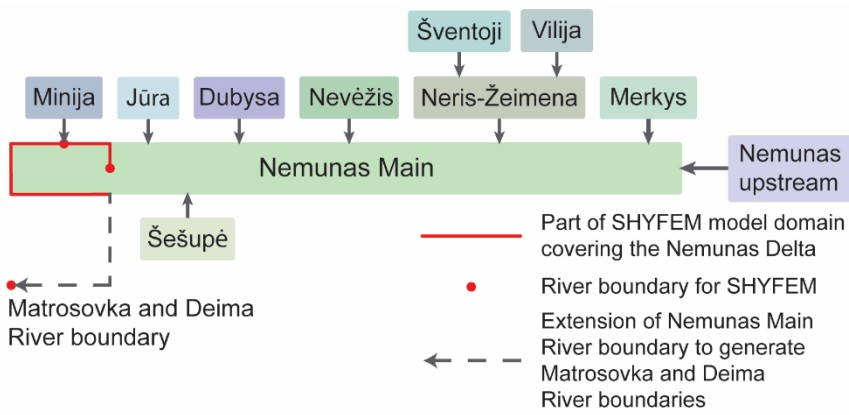

**Figure 2. SWAT models for the Nemunas River watershed and their linkage with SHYFEM through river boundaries.**

### 2.3.2 Hydrodynamic model

Hydrodynamics of the Curonian Lagoon and the south-eastern Baltic Sea were modelled with an open-source shallow water hydrodynamic finite element model SHYFEM (http://www.ismar.cnr.it/shyfem, Umgiesser et al., 2004), which consists of a finite element 3-D hydrodynamic model, a transport and diffusion model, and a radiation transfer model of heat at the water surface. The model resolves the 3-D primitive equations, vertically integrated over each layer, in their formulations with water levels and transports. Horizontal spatial discretization of the model is based on an unstructured triangular grid and carried out using a finite element method, which makes it suitable for applications to coastal systems with complicated geometry and bathymetry.

This model has already been successfully applied to many coastal environments (Umgiesser et al., 2014; De Pascalis et al., 2011; Ferrarin et al., 2013, 2010; Bellafiore and Umgiesser, 2010; Ferrarin and Umgiesser, 2005), as well as validated for the Curonian Lagoon case study in previous works (Mėžinė et al., 2019; Umgiesser et al., 2016; Zemlys et al., 2013; Ferrarin et al., 2008). For this study, SHYFEM was applied in its 2-D version, which is sufficient considering the shallow nature of this lagoon. The computational grid consists of 3292 triangular elements with 1986 nodes having a much finer resolution in the Klaipėda Strait area (Fig. 3). Model produced output data of hydrodynamic properties every 6 hours.

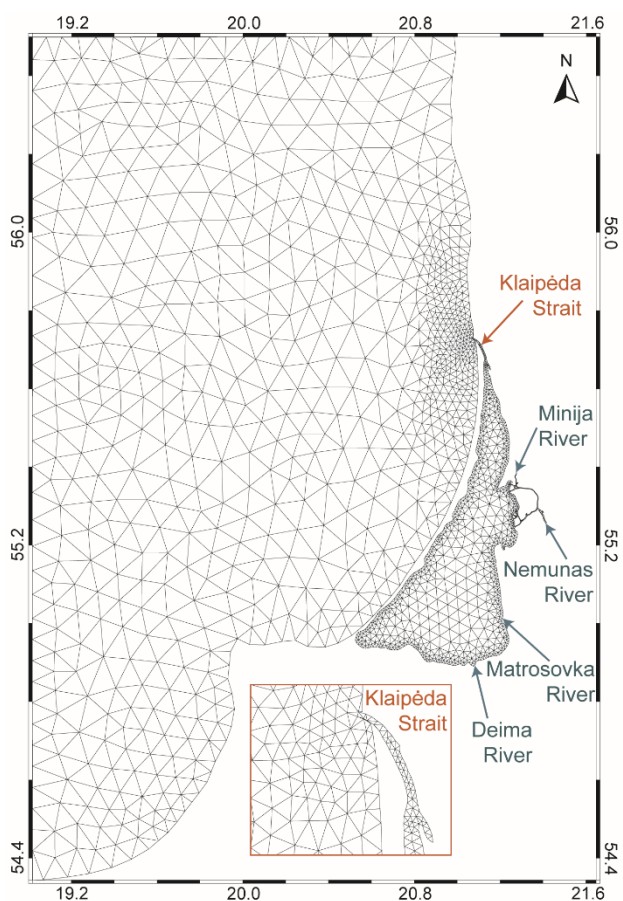

**Figure 3. Computational grid of the hydrodynamic model SHYFEM. Dark blue arrows denote the location of the river boundaries (modelling system connectivity points), where the hydrological model outputs (water temperature and discharge) are used as inputs to the hydrodynamic model.**

### 2.3.3 Coupling of models

The hydrological (SWAT) and hydrodynamic (SHYFEM) models were coupled by taking the daily water temperature and discharge output data from the SWAT model and directly using it as input for the Minija and Nemunas river boundaries in SHYFEM (Fig. 3). However, other two main rivers discharging in the southern part of the lagoon (Matrosovka, branch of the Nemunas, and Deima) are not part of the Nemunas River watershed model, thus the SWAT output data of the Nemunas River were scaled by the ratio between Matrosovka-Nemunas and Deima-Šešupė calculated from the data presented in Jakimavičius (2012). A better representation of hydrodynamic conditions in the Nemunas Delta area is achieved with a hydrodynamic model that simulates the river branches in the delta.

**2.4 Methods of data analysis**

Model results based on four global climate model forcings (Table 1) were averaged, therefore the final analysis was done for three aggregated outputs referring to the historical and projection (RCP4.5 and RCP8.5) scenarios. The analysis periods were divided into historical (1975–2005), short-term (2020–2050) and long-term (2070–2100), and the results were compared by
averaging over these timespans. The period of 1970–1975 was discarded from analysis as it was used for the model spin-up. The main air temperature and precipitation patterns were determined. The parameters of the hydrological model (water discharge) and parameters of the hydrodynamic model (water temperature, salinity, level, residence time, and fluxes through predefined cross sections), as well as ice thickness, were assessed. The analysis was done using standard statistical methods, and computing the percentage of change from the historical period.

**3 Results**

**3.1 Bias correction**

Even though the meteorological data from the global climate models were already downscaled by the regional climate model, there was still some bias remaining in comparison with the local measurement data – underestimation of air temperature and overestimation of precipitation. Therefore, an additional correction was applied (Fig. 4). The correlation between the modelled
and measured air temperature datasets was strong (~0.8), however after bias correction there was an evident improvement of the standard deviation. Similar change is also equivalent to the monthly precipitation.

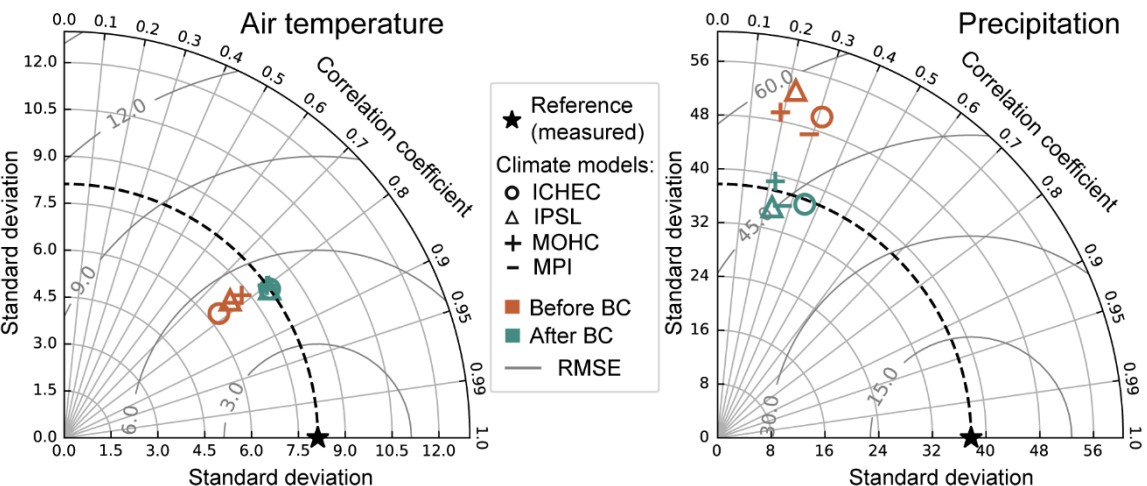

**Figure 4. Taylor diagram showing a statistical comparison of air temperature (daily mean) and precipitation (monthly sum) before and after bias correction (BC) with respect to the reference measurement data. Statistics were computed for the period of 1990-**
**2005.**

## 3.2 Meteorological changes

The bias-corrected statistics of both RCP scenarios show a distinct increase in the projected monthly and annual average precipitation and air temperature (Table 2). The differences between the scenarios in the short-term are smaller than in the long-term by a monthly average of 49% under RCP4.5 and ~123% under RCP8.5. Considering precipitation, the monthly wetness/dryness pattern is similar between the analysis periods and scenarios. The average air temperature in the long-term has the highest increase, especially in the months from November to April. The highest changes of these parameters are projected under RCP8.5 scenario.

**Air temperature (ºC)**

| | | Jan | Feb | Mar | Apr | May | Jun | Jul | Aug | Sep | Oct | Nov | Dec | Year |
|---|---|---|---|---|---|---|---|---|---|---|---|---|---|---|
| Historical (1975-2005) | | -2.6 | -2.6 | 0.3 | 5.3 | 10.7 | 14.6 | 17.9 | 17.5 | 13.2 | 8.1 | 1.8 | -2.3 | 6.8 |
| Short-term (2020-2050) | RCP4.5 | -0.5 | 0.4 | 2.4 | 7.4 | 12.2 | 15.9 | 19.2 | 18.7 | 14.6 | 9.7 | 3.8 | 0.0 | 8.6 |
| | RCP8.5 | -0.6 | 0.5 | 2.8 | 7.5 | 12.4 | 16.1 | 19.2 | 18.8 | 14.6 | 10.0 | 4.2 | -0.2 | 8.8 |
| Long-term (2070-2100) | RCP4.5 | 0.9 | 1.4 | 3.4 | 8.4 | 13.3 | 16.9 | 20.1 | 19.7 | 15.5 | 10.9 | 4.7 | 1.0 | 9.7 |
| | RCP8.5 | 2.9 | 3.4 | 5.0 | 10.0 | 14.7 | 18.1 | 21.8 | 21.5 | 17.6 | 12.4 | 6.7 | 3.3 | 11.5 |

**Precipitation (mm month$^{-1}$)**

| | | Jan | Feb | Mar | Apr | May | Jun | Jul | Aug | Sep | Oct | Nov | Dec | Year |
|---|---|---|---|---|---|---|---|---|---|---|---|---|---|---|
| Historical (1975-2005) | | 42.8 | 36.5 | 43.1 | 36.9 | 55.9 | 87.1 | 57.5 | 57.9 | 48.7 | 63.7 | 39.0 | 48.7 | 617.7 |
| Short-term (2020-2050) | RCP4.5 | 54.4 | 47.6 | 53.7 | 41.1 | 53.1 | 83.4 | 69.9 | 64.7 | 55.6 | 72.7 | 49.9 | 56.4 | 702.5 |
| | RCP8.5 | 53.1 | 46.9 | 50.1 | 43.1 | 61.0 | 83.1 | 75.4 | 69.1 | 54.2 | 71.2 | 50.5 | 58.4 | 716.2 |
| Long-term (2070-2100) | RCP4.5 | 56.9 | 46.5 | 53.7 | 44.8 | 58.8 | 83.3 | 69.2 | 70.0 | 54.0 | 78.2 | 53.6 | 59.3 | 727.3 |
| | RCP8.5 | 72.3 | 52.2 | 65.4 | 49.5 | 67.7 | 93.2 | 73.6 | 77.9 | 60.6 | 85.3 | 62.7 | 70.3 | 830.8 |

**Table 2. Monthly and yearly mean of air temperature and precipitation over the study region derived by averaging four bias-corrected climate models' datasets.**

The difference between land and sea regulate local climate (temperature) and precipitation patterns. Considering the differences between marine and terrestrial parts of the study area, there is a clear pattern of lower air temperatures (on average by ~1–1.5 °C) over the Baltic Sea and the Curonian Lagoon during March through June, while during the rest of the year it is warmer (on average by ~2.2 °C), compared with that of the Nemunas River watershed. During February through August and December months, less precipitation over the marine part (on average by 10 mm month$^{-1}$) is observed, while during the rest of the months it is higher by the same amount compared with the terrestrial area. Therefore, annual precipitation is similar to both areas.

## 3.3 Interactions between domains

The historical discharges of the Nemunas River watershed to the Curonian Lagoon were compared to the projected averaged discharges under RCP4.5 and RCP8.5 climate change scenarios for all seasons (Fig. 5). The most noticeable change is the

projected drastic increase in runoff during winter, which will almost double in the long-term period under both RCPs. This result is a combination of increased projected average precipitation and temperatures, which will likely result in warmer winters with reduced snow cover throughout the basin. The increased liquid precipitation in winter will likely lead to an increased soil erosion and nutrient wash-off, when the soil is bare and exposed to the elements, such as rain and wind.

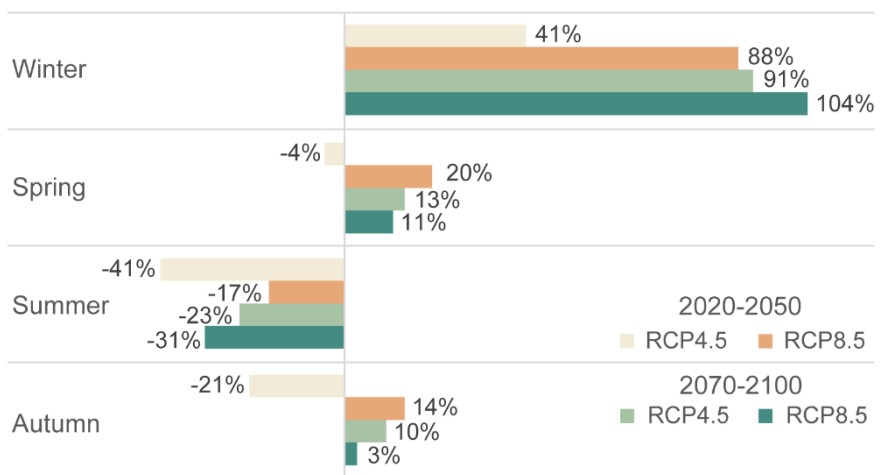

**Figure 5. The percentage of change of seasonal average discharge from the Nemunas River for the RCP4.5 and RCP8.5 scenario runs during the short-term (2020–2050) and long-term periods (2070–2100), compared with the historical period (1975–2005).**

In contrast, a decrease in average summer outflow is projected under both RCP scenarios, despite an increase in precipitation across the region. The increased average temperatures in summer months will impact the evapotranspiration (ET) over the basin, where an increase in both potential ET and ET is projected, which will lead to lower discharges at the stream gages

during this period. Stable and mostly unchanged Nemunas River discharge will continue to supply the lagoon over the rest of the seasons, in spring and autumn.

Noteworthy are the differences between the RCP scenarios. We see a stronger shift in the increased winter discharges in the RCP8.5 scenario in the short-term period, and a stronger shift towards lesser summer flows in the RCP4.5 scenario (Fig. 5). The RCP8.5 in the short-term can be considered as "wetter" scenario, whereas RCP4.5 is "dryer". In the long-term, however,

both RCPs simulate similar outcomes with slightly different magnitudes.

The interactions between the study domains can be defined by the water fluxes from the main discharging rivers into the Curonian Lagoon, eventually into the Baltic Sea (water outflow), and vice versa (water inflow). The volume of inflowing water is much lower than that of the outflowing (Fig. 6), especially in the northern part of the Curonian Lagoon, meaning that the outflow is already dominating and this study shows that it will prevail and even increase in the analysed continuum system.

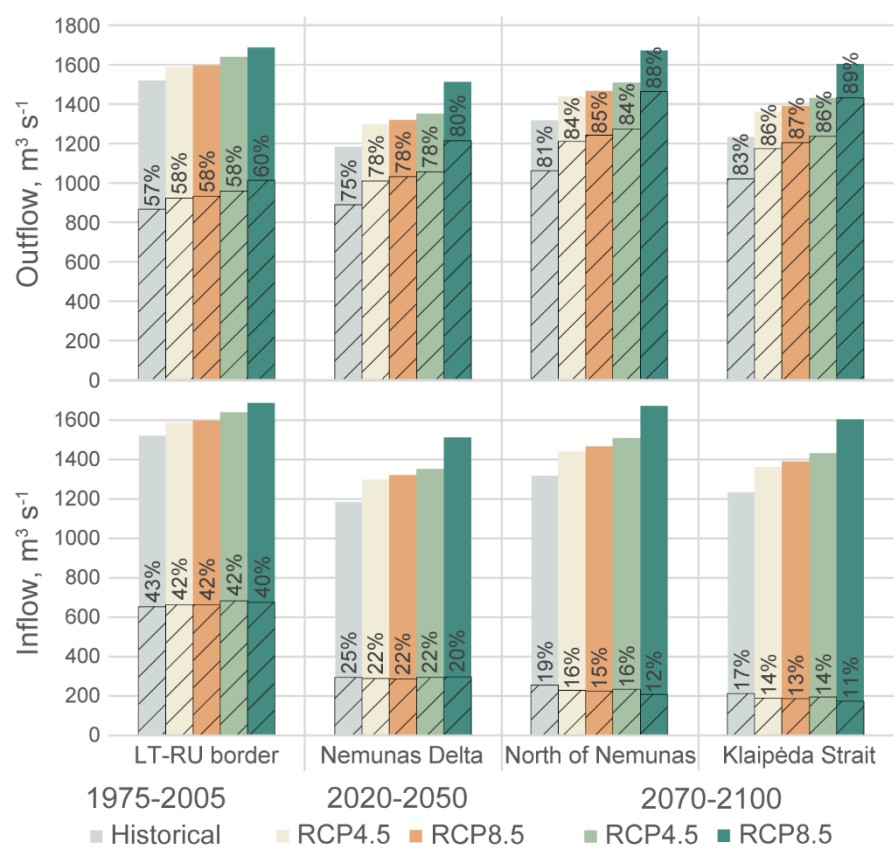

**Figure 6. Average outflowing and inflowing water volume (hatched pattern) including its percentage of the absolute flux (solid pattern) of each analysis period – historical (1975–2005), short-term (2020–2050), and long-term (2070–2100) under RCP4.5 and RCP8.5 scenarios. The locations of cross-sections are shown in Fig. 7.**

Comparing the simulation results of the scenario runs (short- and long-term) with the historical period, it is evident that the water inflow in the northern part of the lagoon (cross-sections 1 and 2 in Fig. 7) is projected to decrease. The change of water inflow in the centre of the lagoon (cross-sections 3 and 4 in Fig. 7) is very small and can be considered negligible. The outflowing water flux is projected to increase and is related to the growing outflow from the Nemunas River and will increase along the pathway towards the Baltic Sea (cross-sections 1, 2, and 3 in Fig. 7). In the southern part of the lagoon, near the Lithuanian–Russian border, the changes in water fluxes are less evident, due to the fact that water circulation in this area is mostly determined by the wind induced currents.

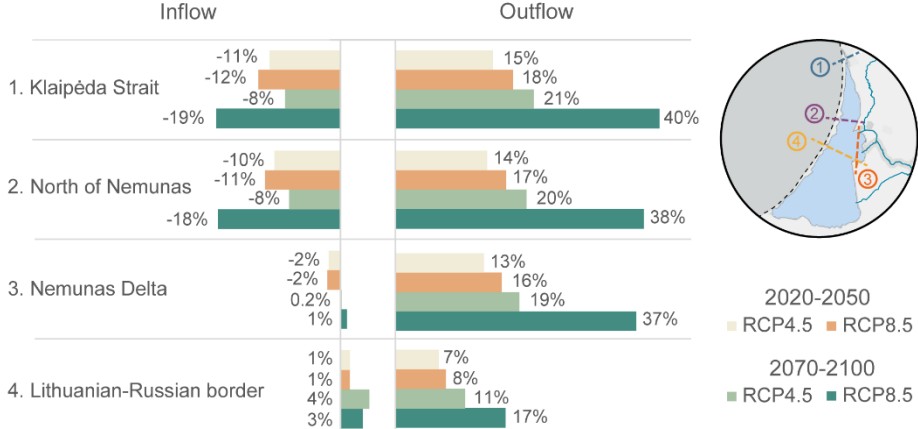

**Figure 7. The percentage of change of projected (under RCP4.5 and RCP8.5 scenarios) inflowing and outflowing water flux from the historical period (1975–2005). The map insert shows the locations of the predefined cross-sections: 1 – Klaipėda Strait, 2 – North of Nemunas, 3 – Nemunas Delta, 4 – Lithuanian–Russian border. Outflow is from south to north (sections 1,2,4), and from east to west (section 3), inflow - vice versa.**

The change of water flow through all cross-sections display a similar pattern in all scenarios, the water flux changes from the historical are very similar in each analysed cross-section and for each flow type. For both the outflow and inflow the highest change from the historical period is observed in the long-term under RCP8.5 scenario.

Looking at the water exchange seasonal dynamics, we can see that the highest inflowing water deviations from the historical period are predicted during winter and spring, being lower by 10–28% (Table 3). The water exchange is expected to increase in summer by 12–22% while in the autumn the changes are very small and similar throughout the analysis periods and scenarios.

|  |  | 1975-2005 | 2020-2050 | | 2070-2100 | |
|---|---|---|---|---|---|---|
|  |  | Historical | RCP4.5 | RCP8.5 | RCP4.5 | RCP8.5 |
| OUTFLOW | Winter | 1017 | 1291 (27%) | 1324 (30%) | 1388 (36%) | 1663 (63%) |
|  | Spring | 1046 | 1210 (15%) | 1213 (15%) | 1214 (16%) | 1356 (29%) |
|  | Summer | 858 | 814 (-5%) | 850 (-1%) | 856 (0.3%) | 923 (8%) |
|  | Autumn | 924 | 1007 (9%) | 1030 (12%) | 1076 (17%) | 1190 (29%) |
| INFLOW | Winter | 452 | 387 (-19%) | 383 (-20%) | 380 (-21%) | 357 (-28%) |
|  | Spring | 246 | 225 (-13%) | 220 (-16%) | 233 (-10%) | 225 (-14%) |
|  | Summer | 277 | 309 (13%) | 308 (12%) | 331 (22%) | 315 (14%) |
|  | Autumn | 439 | 449 (2%) | 450 (2%) | 461 (4%) | 456 (2%) |

**Table 3. Average seasonal water flux in $m^3$ $s^{-1}$ over historical (1975–2005), short-term (2020–2050), and long-term (2070–2100) periods under RCP4.5 and RCP8.5 scenarios. Percentage in the brackets shows the change compared with the historical period. The values are averaged over four cross-sections in the Curonian Lagoon: Klaipėda Strait, North of Nemunas, Nemunas Delta, and Lithuanian Russian border (see map insert in Fig. 7).**

The changes of the outflow during winter, spring, and autumn are expected to increase (Table 3). In the short-term, the changes between the scenarios are similar, while in the long-term, the change increases nearly twice under RCP8.5, compared with RCP4.5. During summer the outflow changes are negligible, apart from the long-term scenario under RCP8.5 (-8%).

## 3.4 Water residence time

The water residence time (WRT) in the Curonian Lagoon generally is likely to have a decreasing tendency, apart from the summer season, when it slightly increases (Table 7). In the northern part of the lagoon WRT is always shorter, while the southern part has higher values, because the water movement in this area is more driven by wind than by the outflowing rivers (e.g., Nemunas). There is no apparent difference between the climate scenarios in the short-term. The highest decrease, compared with the historical period, is observed in the long-term under RCP8.5 scenario.

| | | 1975-2005 Historical | 2020-2050 | | 2070-2100 | |
|---|---|---|---|---|---|---|
| | | | RCP45 | RCP85 | RCP45 | RCP85 |
| North | Winter | 61 | 49 (-21%) | 48 (-21%) | 46 (-24%) | 41 (-32%) |
| | Spring | 44 | 39 (-11%) | 40 (-9%) | 41 (-7%) | 39 (-12%) |
| | Summer | 59 | 63 (7%) | 63 (6%) | 66 (11%) | 61 (4%) |
| | Autumn | 67 | 66 (-1%) | 66 (-1%) | 67 (0.03%) | 62 (-7%) |
| South | Winter | 174 | 123 (-29%) | 119 (-32%) | 112 (-36%) | 88 (-49%) |
| | Spring | 122 | 96 (-22%) | 99 (-19%) | 101 (-17%) | 94 (-23%) |
| | Summer | 179 | 196 (9%) | 199 (11%) | 217 (21%) | 192 (7%) |
| | Autumn | 180 | 178 (-1%) | 178 (-1%) | 176 (-2%) | 158 (-12%) |
| Total | Winter | 127 | 95 (-25%) | 93 (-27%) | 88 (-30%) | 73 (-43%) |
| | Spring | 94 | 76 (-18%) | 79 (-16%) | 80 (-14%) | 75 (-20%) |
| | Summer | 129 | 140 (8%) | 140 (8%) | 147 (14%) | 134 (4%) |
| | Autumn | 135 | 133 (-1%) | 133 (-1%) | 133 (-2%) | 120 (-11%) |

**Table 7. Seasonal water residence time (in days) in the northern, southern, and total lagoon area under RCP4.5 and RCP8.5 scenarios, averaged over historical (1975–2005), short-term (2020–2050), and long-term (2070–2100) periods. Percentage in the brackets shows the change compared with the historical period.**

The highest WRT in the Curonian Lagoon is during summer and autumn when the discharge from the surrounding rivers decreases. Seasonal changes compared with the WRT during the historical period revealed that the highest decrease is observed during winter, followed by spring. During summer WRT slightly increases under both RCP scenarios and both analysis periods, while in autumn the changes are negligible, apart from RCP8.5 scenario in the long-term, when the WRT decreases by ~10%, compared to the historical period.

## 3.5 Salinity

The modelling results show that the Baltic Sea saltwater intrusions into the Curonian Lagoon will likely decrease in the future. During the historical period 48 days year[-1] of saltwater intrusions were observed, i.e., the average yearly number of days when

salinity exceeds the 2 g kg⁻¹ threshold near Juodkrantė, which is approximately 20 km below the lagoon-sea connection. In the short-term, this number will decrease by 60% under RCP4.5 scenario (19 days year⁻¹) and 65% under RCP8.5 scenario (17

days year⁻¹) compared with the historical period. While in the long-term, it will decrease by ~67% under RCP4.5 (16 days year⁻¹) and 83% under RCP8.5 scenario (8 days year⁻¹).

Seasonally, the highest change of saltwater intrusion days is observed during winter and especially spring, when the outflow of the lagoon is dominating and the salinity values in Juodkrantė do not exceed 2 g kg⁻¹ (right side of Fig. 8). Saltwater intrusions during summer season have a larger decrease under RCP8.5 scenario than RCP4.5, which does not have a distinct

pattern in the future. In autumn, the changes in the short-term are similar under both scenarios ~27% decrease compared with the historical period, and it continues to decline in the long-term as well.

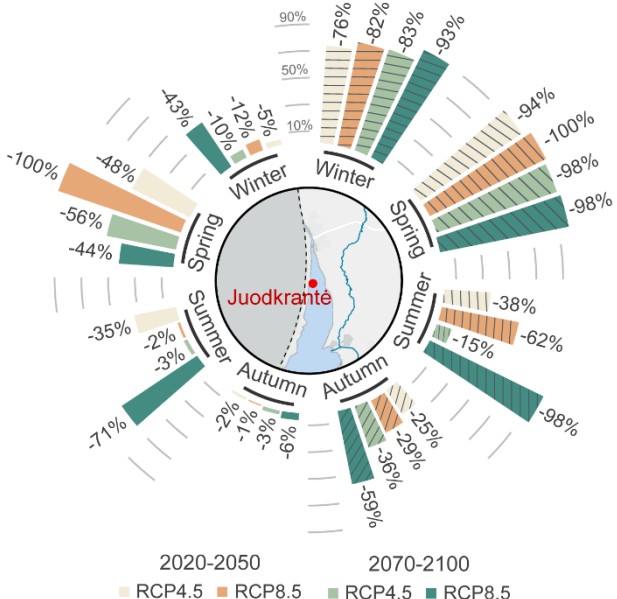

**Figure 8. The percentage of change of seasonal average salinity of saltwater intrusions (solid pattern – left) and number of days when salinity exceeded the 2 g kg⁻¹ threshold (hatched pattern – right) in Juodkrantė, approximately 20 km below the strait**
**connecting the Baltic Sea and the Curonian Lagoon. Values denote the percentage of difference of the scenario runs compared with the historical period (1975–2005).**

The severity of the saltwater intrusions is also predicted to decrease (left side of Fig. 8). The pattern is similar to that of the number of days of saltwater intrusions. Due to the increased outflow from the lagoon during spring, the changes are the highest in the same period.

Salinity in the south-eastern Baltic Sea show a decreasing pattern throughout the analysis periods. Under RCP4.5 scenario, there is a small salt content decrease projected – 5–6% less in the short-term and 11–18% in the long-term (Table 4), compared with the historical period. Difference between the RCP scenarios are only apparent in the long-term, where the change is projected to be higher under RCP8.5 scenario. In the Baltic Sea there is no difference of average salinity throughout the seasons.

| 1975-2005 | 2020-2050 | 2070-2100 |
|---|---|---|

|  |  | Historical | RCP4.5 | RCP8.5 | RCP4.5 | RCP8.5 |
|---|---|---|---|---|---|---|
| Coastal waters | Winter | 6.6 | 5.9 (-11%) | 5.8 (-13%) | 5.4 (-18%) | 4.8 (-27%) |
| | Spring | 5.4 | 4.9 (-11%) | 4.8 (-12%) | 4.4 (-18%) | 4.0 (-26%) |
| | Summer | 5.4 | 5.1 (-4%) | 5.0 (-7%) | 4.8 (-12%) | 4.3 (-20%) |
| | Autumn | 6.3 | 5.9 (-6%) | 5.8 (-8%) | 5.5 (-13%) | 5.0 (-21%) |
| Baltic Sea | Winter | 7.8 | 7.4 (-5%) | 7.3 (-6%) | 6.9 (-11%) | 6.4 (-18%) |
| | Spring | 7.7 | 7.4 (-5%) | 7.3 (-6%) | 6.9 (-11%) | 6.3 (-18%) |
| | Summer | 7.8 | 7.4 (-5%) | 7.3 (-6%) | 7.0 (-11%) | 6.4 (-18%) |
| | Autumn | 7.8 | 7.4 (-5%) | 7.3 (-6%) | 7.0 (-11%) | 6.4 (-18%) |

**Table 4. Average seasonal salinity in the Baltic Sea and along the south-eastern Baltic coast. Percentage in the brackets shows the**
**change of the scenario runs compared with the historical period (1975–2005).**

Salinity in the coastal area is slightly lower compared to that of the Baltic Sea. However, here the difference of seasons is more apparent, having a higher decrease of salinity values during winter and spring when the discharge of freshwater from the Curonian Lagoon increases. In the coastal waters the highest change, compared with the historical period, is also projected to be under RCP8.5 scenario. Notably, the highest decrease is projected northward of Klaipėda Strait (lagoon outlet), while
southward the change is lower and salinity values are similar to that of south-eastern Baltic Sea.

### 3.6 Water temperature

The water temperature in the Curonian Lagoon is relatively homogeneous throughout its area with slightly warmer water (by on average 0.4 °C) in the northern part of the lagoon. Considering the seasonal dynamics, water temperature is projected to increase the most during winter, by ~60% in the short-term and more than twice in the long-term (Table 5). During spring and
autumn, the increase is lower – ~17% in the short-term and 19–27% in the long-term. The lowest increase is projected during summer season – 5–6% in the short-term and 9–13% in the long-term. However, considering the water temperature increase in °C, we can see that it in the short-term it is similar between the seasons ranging from on average 0.9 °C (in winter) to 1.4 °C (in spring), while in the long-term the change is higher – 1.9 °C in winter and spring, 2.2 °C in summer, and .3°C in autumn. The more apparent difference between the RCP scenarios is observed during the long-term, where RCP8.5 displays a higher
temperature increase.

|  |  | 1975-2005 | 2020-2050 | | 2070-2100 | |
|---|---|---|---|---|---|---|
|  |  | Historical | RCP4.5 | RCP8.5 | RCP4.5 | RCP8.5 |
| Lagoon | Winter | 1.5 | 2.5 (60%) | 2.5 (61%) | 3.1 (102%) | 3.9 (151%) |
| | Spring | 8.3 | 9.6 (16%) | 9.8 (18%) | 9.9 (19%) | 10.5 (27%) |
| | Summer | 19.8 | 20.8 (5%) | 20.9 (6%) | 21.6 (9%) | 22.3 (13%) |

| | | Historical | RCP4.5 | RCP8.5 | RCP4.5 | RCP8.5 |
|---|---|---|---|---|---|---|
| | Autumn | 10.5 | 11.7 (11%) | 12.0 (14%) | 13.1 (24%) | 13.8 (32%) |
| Coastal waters | Winter | 2.6 | 3.7 (42%) | 3.7 (42%) | 4.4 (72%) | 5.9 (129%) |
| | Spring | 5.4 | 6.7 (24%) | 6.8 (26%) | 7.0 (30%) | 8.4 (56%) |
| | Summer | 16.0 | 17.0 (7%) | 17.2 (8%) | 17.7 (11%) | 19.2 (20%) |
| | Autumn | 12.0 | 13.1 (9%) | 13.4 (11%) | 14.4 (20%) | 15.8 (31%) |
| Baltic Sea | Winter | 4.2 | 5.2 (24%) | 5.3 (26%) | 5.9 (41%) | 7.2 (71%) |
| | Spring | 3.4 | 4.5 (30%) | 4.6 (33%) | 5.1 (50%) | 6.3 (85%) |
| | Summer | 7.4 | 8.5 (14%) | 8.6 (16%) | 9.2 (23%) | 10.4 (40%) |
| | Autumn | 8.3 | 9.3 (12%) | 9.4 (13%) | 10.1 (21%) | 11.3 (35%) |

**Table 5. Average seasonal water temperature (in °C) in the Curonian Lagoon and Baltic Sea. Percentage in the brackets shows the change of the scenario runs compared with the historical period (1975–2005).**

In general, in the Baltic Sea the highest change of water temperature is projected in the long-term than in the short-term (Table 5). Seasonally, the highest increase is estimated in winter and spring during both analysis periods, of which the highest change is under RCP8.5 scenario, compared with the historical period. The smallest change throughout the analysis periods is projected in summer and autumn. There is no apparent difference between the RCP scenarios in the short-term, contrary to the long-term, when the RCP8.5 scenario displays a higher water temperature increase.

The seasonal dynamics in the coastal area (nearshore temperature) is very similar to the whole south-eastern Baltic Sea. Although, the water temperature is slightly lower during the winter season, on average by 1.5 °C, and higher during other seasons, on average by 4.9 °C. The changes between the analysis periods and RCPs scenarios comparing with the historical period are higher than that of the south-eastern Baltic Sea.

### 3.7 Water level

The water level in the Curonian Lagoon usually varies between -0.2–0 m during the historical period. In the short-term projections, the model estimates a water level increase by 17–20 cm compared with the historical period (Table 6). The highest increase can be estimated in the long-term, when the water level in the lagoon increases by 32 cm under RCP4.5 and 41 cm under RCP8.5 scenario, compared with the historical period. For the latter, the highest increase is observed in the Nemunas Delta and northern part of the lagoon.

| | 1975-2005 | 2020-2050 | | 2070-2100 | |
|---|---|---|---|---|---|
| | Historical | RCP4.5 | RCP8.5 | RCP4.5 | RCP8.5 |
| Baltic Sea | -0.15 | 0.01 | 0.03 | 0.17 | 0.23 |
| Klaipėda Strait | -0.13 | 0.04 | 0.06 | 0.19 | 0.26 |
| Curonian Lagoon | -0.07 | 0.10 | 0.13 | 0.25 | 0.34 |

**Table 6. The average water level during the historical (1975–2005) and the short- (2020–2050) and long-term (2070–2100) periods under RCP4.5 and RCP8.5 scenarios. Data were averaged over the south-eastern Baltic Sea, Klaipėda Strait, and the Curonian Lagoon.**

The water level in the Baltic Sea, as well as the coastal area, is homogeneous and most of the time staying below the sea level during the historical period. The water level in the sea is expected to rise by 16–19 cm in the short-term and by 32–39 cm in the long-term, compared with the historical period (Table 6). This change is one half of that in the Curonian Lagoon. At the domains' connective area – Klaipėda Strait, the water level corresponds mostly to the dynamics of it in the sea.

## 3.8 Ice thickness

Ice thickness in the Curonian Lagoon is projected to decrease steadily. The average maximum ice thickness and ice season duration during the short-term period can decrease by 25% compared with the historical period. Additionally, there is no apparent difference between the RCP scenarios (Fig. 9). Adverse case is with long-term projections, for which the average maximum ice thickness under RCP4.5 scenario is estimated to decrease by half and by 70% under RCP8.5 scenario, compared with the historical period. The ice duration is as well projected to continue becoming shorter by 35% under RCP4.5 and 67% under RCP8.5 scenario, compared with its duration during the historical period. These changes indicate a possible severe modification of ice regime in the lagoon, leading to winter seasons with a more likely unstable thin ice cover.

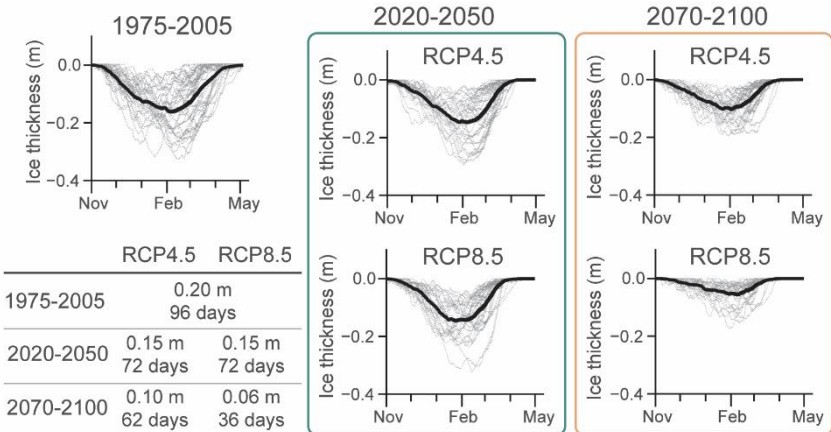

| | RCP4.5 | RCP8.5 |
|---|---|---|
| 1975-2005 | 0.20 m | |
| | 96 days | |
| 2020-2050 | 0.15 m | 0.15 m |
| | 72 days | 72 days |
| 2070-2100 | 0.10 m | 0.06 m |
| | 62 days | 36 days |

**Figure 9. Average ice thickness in the Curonian Lagoon. Black line indicates the average ice thickness of each analysis period – historical (1975–2005), short-term (2020–2050), and long-term (2070–2100) under RCP4.5 and RCP8.5 scenarios. Table shows the average maximum ice thickness (m) and ice season duration (days) of the scenario runs compared with the historical period.**

## 4 Discussion

The results above give an indication on the variation of the physical parameters that the Nemunas River watershed, Curonian Lagoon, and south-eastern part of the Baltic Sea will be subjected to. All parameters indicate a change, more or less strong, until the end of this century. These changes will be discussed hereafter.

### 4.1 Local climate insights

The results of this study have been achieved by applying a coupled hydrological-hydrodynamic modelling system, which was run with input data from regional climate models that have been downscaled from global models. The quality of the regional climate model output was generally good, but a comparison with local observed data showed the need to ultimately correct the downscaled data by applying a bias correction. This was needed because some variables (precipitation suffices as one example) were far from being well produced by the regional climate models (Tapiador et al., 2019).

As summarized in the BACC assessments – the Baltic Sea region is warming and it will continue to warm throughout the 21st century (Meier et al., 2022a). The European Environment Agency (EEA) has reported that climate models project a future increase in average temperature across Europe, with the largest warming expected in the eastern part of the continent. Accordingly, our study results show a considerable increase in the projected air temperatures as well (Table 8). These changes will likely result in warmer conditions during winters, less snow and ice cover as well as higher river discharge, while the increase of air temperature during other seasons might have a severe impact on the crop production, increase the occurrence of hydrological and/or agricultural droughts, and impact changes in the crop calendar of the region. During summer months, rising temperatures could exacerbate water scarcity and increase demand for irrigation (Christensen, 2022). This could have significant implications for agriculture, as well as for water management and infrastructure planning in the region.

| Parameter | Domain | 2020-2050 | | 2070-2100 | |
|---|---|---|---|---|---|
| | | RCP4.5 | RCP8.5 | RCP4.5 | RCP8.5 |
| Air temperature | Marine area | 21% | 23% | 34% | 57% |
| | Terrestrial area | 33% | 35% | 51% | 81% |
| Precipitation | Marine area | 14% | 15% | 18% | 34% |
| | Terrestrial area | 14% | 16% | 18% | 35% |
| Water inflow | Nemunas River to Lagoon | 13% | 16% | 19% | 37% |
| Water outflow | Lagoon to sea | 15% | 18% | 21% | 40% |
| Salinity | Coastal area | -8% | -10% | -15% | -24% |
| | Sea | -5% | -6% | -11% | -18% |
| Saltwater intrusions (No of days year$^{-1}$) | Juodkrantė | -30% | -34% | -34% | -51% |
| Water temperature | Lagoon | 11% | 13% | 19% | 26% |
| | Coastal area | 13% | 14% | 21% | 37% |
| | Sea | 17% | 19% | 30% | 50% |
| Water residence time | Northern part of the lagoon | -6% | -6% | -5% | -12% |
| | Southern part of the lagoon | -10% | -9% | -8% | -19% |
| | Total lagoon area | -8% | -8% | -8% | -17% |

| | | | | | |
|---|---|---|---|---|---|
| Maximum ice thickness | Lagoon | -25% | -25% | -50% | -70% |
| Ice season duration | Lagoon | -25% | -25% | -35% | -63% |

**Table 8. Summary of the percentage change of projected average annual changes (under RCP4.5 and RCP8.5 scenarios) during short-term (2020–2050) and long-term (2070–2100) periods compared with the historical period (1975–2005).**

The EEA also notes that changes in precipitation patterns, including a tendency towards drier summers and wetter winters, are

projected in many regions of Europe (EEA, 2021). Climate models project an increase in annual precipitation in many areas of eastern Europe, but with a strong seasonal shift towards more winter precipitation and less summer precipitation (Jacob et al., 2014). Our results show that by the end of the century an increase of the total annual precipitation by 18 to 34% (~100–200 mm) according to RCP4.5 and RCP 8.5 scenarios respectively, is projected. In comparison with a neighbour country, this is considerably higher than the projected precipitation for the Latvian case, where the total annual increase was estimated to

be 13 to 16% (~80–100 mm) under the same scenarios (Avotniece et al., 2017). This can lead to increased runoff and river discharge, especially during the winter months, potentially increasing the risk of flooding, while summer months could become drier and more prone to drought.

**4.2 Water flow**

The driving force of changes in the lagoon are the boundaries – the Nemunas River inputs and the influence of the Baltic Sea

through the Klaipėda Strait. Historically every year on average Nemunas River supplies three times the volume of water to the lagoon (Žilinskas et al., 2012), which is subject to change in the future. The projections indicate a 13–16% (in the short-term) and 19–37% (in the long-term) increase in water discharge from the river, which coincides with the projected precipitation increase (Table 8). This projected change alone will strongly influence the hydrodynamic conditions of the lagoon.

Due to the enhanced Nemunas discharge, the fluxes from the Curonian Lagoon into the Baltic Sea will also increase. The

outflow to the sea is already dominating, especially in the northern part of the lagoon and it will remain so in the future. The water flow will increase in all its path towards the Baltic Sea by 13–18% in the short-term and by ~20% under RCP4.5 scenario in the long-term, while the changes under RCP8.5 scenario can be expected to increase by up to 40% by the end of the century, compared with the historical period (Table 8).

The volume of discharging river water into the Baltic Sea is projected to be dependent on the season and the region (Graham,

2004; Meier et al., 2022a). With this, the increase of riverine nutrient loads is also anticipated (Pihlainen et al., 2020), likewise suggesting that "land based nutrient management will have greater effect on nutrient loads than greenhouse gas emissions" (Climate Change in the Baltic Sea: 2021 Fact Sheet).

The seasonal river discharge changes in the Baltic Sea correspond with our study noting that a more likely increase in winter and decrease in summer can be expected by the end of this century (Donnelly et al., 2014). The magnitude of changes of

freshwater inflow within different subregions of the Baltic Sea has a considerable variation, i.e., an increased outflow in the northern part of the Baltic Sea and a decreased in the southern part (Graham, 2004). Although the combined uncertainties from

the climate models, their bias correction, and applied models make it difficult to draw conclusions about the magnitude of change (Donnelly et al., 2014).

The most alarming are the projected hydrologic changes in the Nemunas River watershed during the winter season in the long-term period. As supported by other studies (Čerkasova et al., 2021; Stonevičius et al., 2017), both RCPs point toward a drastic increase in winter discharges, which might lead to severe flooding of the delta region, which suffers from annual floods of different magnitude (Valiuškevičius et al., 2018). Currently these flood events occur in spring, but will likely occur in winter in the future, and with increased severity, which will pose a serious concern to the local population and stakeholders. The currently functioning polder system in the Nemunas River delta (Lesutienė et al., 2022) region might not withstand such changes.

## 4.3 Water residence time in the lagoon

Water residence time (WRT) is generally decreasing with climate change (Table 8), although the decline is small in the short-term and in the long-term under RCP4.5 scenario (5–10% less). A general decrease can be found in the whole area of the lagoon in the long-term under RCP8.5 (17%). As in the case with saltwater intrusions, this is due to the increased discharge from the Nemunas River into the lagoon and further outflowing to the sea, where WRT are less varying in the north, whereas it decreased sensibly in the south.

When looking at seasonal changes, WRT becomes lower in summer due to decreasing river discharge. Comparing the changes from the historical period it revealed that the highest decrease is observed during winter and spring, presumably due to the increased outflowing water fluxes and change of the ice cover regime.

## 4.4 Water level dynamics

Water levels are the main drivers, together with the Nemunas discharge, of the exchanges through the Klaipėda Strait. In the strait and particularly in the Curonian Lagoon water levels are higher than in the Baltic Sea, which leads to an increase in outflow periods through the strait. Since 1960 until 2008, the average measured water level in the Curonian Lagoon has been rising at a rate of approximately 3.0 mm year$^{-1}$, leading to an increase by 18 cm during this period (Čepienė et al., 2022, and references therein). Our modelling results showed that in the future, water level is projected to increase by 16–20 cm in the short-term and by 32–41 cm in the long-term, compared with the historical period.

Since the projected water level will be above the mean sea level, it could affect the infrastructure and residents living in the coastal area leading to significant social and economic damages (Vousdoukas et al., 2020). The low elevation wetland areas covering the southern and eastern part of the lagoon (especially Nemunas Delta area) are more likely of getting flooded. However, the Klaipėda city and the Curonian Spit are located higher and the mean water level rise should not permanently affect these areas. It is worth mentioning, that the hydrodynamic model has a fixed boundary, thus the increase could be overestimated especially considering the role of coastal wetlands in mediation of global change induced water level changes.

## 4.5 Salinity dynamics

When looking at the saltwater intrusion events, it can be clearly seen that the intrusion periods are decreasing and the outflow from the lagoon becomes dominant, simply because the outflowing water fluxes will stop the salty seawater from entering the Curonian Lagoon. This is clearly evident for both climate change scenarios – periods where values higher than the threshold of 2 g kg$^{-1}$ in Juodkrantė drop by 30–50% compared with the historical period (Table 8). This will lead to some alteration of the ecosystem in the northern part of the Curonian lagoon, which is now considered as oligohaline.

As seen and suggested from other Baltic Sea salinity projection studies, the salinity in the Baltic does not have a distinct pattern of changing in the future and the projections have a lot of uncertainties (Lehmann et al., 2022; Meier et al., 2022b). Our study of salinity changes in the south-eastern Baltic Sea demonstrates a decreasing tendency, especially in the coastal area, northward from the lagoon outlet, where the sea water is diluted by discharging freshwater from the lagoon (Table 8). However, the scenarios that we used to run the hydrodynamic model did not consider the global sea level rise, which according to Meier et al. (2021) could more or less outweigh the projected increase of the river runoff into the Baltic Sea, due to the higher volume of inflow from the adjacent North Sea.

## 4.6 Water temperature dynamics

The mean temperature of the water column in the Klaipėda Strait is less sensitive to changes, while the lagoon is expected to see an increase in temperature under both RCP scenarios (Table 8). Long-term projections under RCP8.5 show the highest increases. The south-eastern Baltic Sea also shows increasing temperature patterns, with the highest changes occurring during winter and spring. Coastal areas have similar dynamics, but with higher changes compared with the historical period. Coastal average water temperature is lower by 1.5 °C in winter and higher by 4.9 °C during the rest of the year compared to the south-eastern Baltic Sea.

Our study supports the findings of projected increasing water temperature in the Baltic Sea (Meier et al., 2022b). In this article we focused only on the average water temperature of the water column, however, there are many studies revealing the warming patterns of the sea surface temperature in the Baltic (Dutheil et al., 2022; Zhu et al., 2022).

## 4.7 Ice in the lagoon

Ice thickness in the lagoon will become thinner for a longer period during the ice season compared to the historical period. The projected changes of 25% decreased maximum ice thickness and duration in the short-term, and 50–70% maximum thickness and 35-63% ice duration decrease in the long-term (Table 8), could presume leading to severe alteration of the ice season regime in the future. This pattern corresponds to the trends in the southern Baltic lagoons (Girjatowicz and Świątek, 2021) and gulf of Riga (Siitam et al., 2017; Kļaviņš et al., 2016;), as well as Baltic Sea itself (Luomaranta et al., 2014).

Ice in the Curonian Lagoon has become a trending topic in the recent years, due to its already evident changes as seen from the historical observations (Jakimavičius et al., 2019), remote sensing (Kozlov et al., 2020; Idzelytė et al., 2019), as well as

modelling studies of future projections (Idzelytė and Umgiesser, 2021; Jakimavičius et al., 2019). The decreasing ice
505 parameters impact on the hydrodynamics of the lagoon suggest a possible increase of water flow and saltwater intrusions, as well as shorter WRT during the ice cover season (Idzelytė et al., 2020; Umgiesser et al., 2016). Although, the role of ice for controlling WRT is far less than the increased fluxes from the discharging rivers.

## 4.8 Impacts on the ecosystem structure and functions

Despite clear trends for physical indicators, the actual consequences of climate change for the functioning of the Curonian
510 Lagoon ecosystem are much more complicated to predict. The foreseen minor salinity changes in the mostly freshwater ecosystem are unlikely to cause significant shifts in the ecosystem functioning even in the northern part of the lagoon. However, some decrease in distribution areas of some estuarine species (mysids, barnacles) could be foreseen. According to the latest combined genetic and hydraulic modelling study (Fernández-Alías et al., 2022), the connectivity between the Curonian lagoon and coastal Baltic sea fish population could be altered due to the decrease in marine intrusions during the spring spawning
515 period common to the most cyprinid species common for the Curonian lagoon.

A recent analysis (Ivanauskas et al., 2022) indicated a statistical relationship between WRT and salinity and both catches and populations of main commercial fish populations also stating that predicted decrease in salinity and increasing temperatures are favourable for main commercial fish species. This is in stark contrast to the predictions for the Baltic Sea fishery, where catches are expected to be decreasing in quantity, and especially in the quality (Climate Change in the Baltic Sea, 2021). This
520 could be explained by the fact that commercial catches in the Curonian Lagoon already are dominated by eutrophication tolerant species like bream and pikeperch, which are also known to prefer higher water temperatures. However, projected ice cover decrease will inevitably negatively affect the winter ice fishing practices in the lagoon.

The latest studies, revealing the mechanisms of nutrient cycling and eutrophication processes (Bartoli et al., 2018), point toward the expected reduction of the role of the Curonian Lagoon as a coastal filter. The shortening of the ice cover period
525 will reduce the system denitrification capacity while the decrease of WRT will lead to substantial reduction of nutrient and organic matter retention due to the decreased burial and absorption rates. So far, the only difference is the more favourable conditions for the cage aquaculture in the lagoon, where the decreased WRT and increased flushing in combination with modern practices of multi trophic aquaculture (e.g. floating vegetable gardens) could remove some of the constraints preventing the development of cage aquaculture at present.

## 4.9. Study limitations and uncertainty

One potential limitation of the study is that the future scenarios only account for changes in climatic conditions, without taking into consideration the evolution of the drainage basin or the changes in the lagoon and the port area due to socio-economic development. These factors are expected to impact land use, water requirements, surface runoff, and aquifer functioning, as well as hydrodynamic conditions in the lagoon-sea interface. Considering the political situation of the neighbouring countries,

socio-economic changes in this region are extremely difficult to predict. While this means that the predictions may not precisely reflect what will happen, the study still provides a valuable reference point for understanding the projected changes in the region. It can also serve as a necessary starting point for the development of more comprehensive socio-economic predictive models.

It is worth mentioning that the analysed physical changes in the presented continuum system were assessed in the scope of climate change under RCP4.5 and RCP8.5 scenarios; the RCP2.6 mitigation scenario (van Vuuren et al., 2011) was not covered in this study. Whereas tendencies could be identified in our analysis, it was yet not possible to discuss the uncertainty of the results achieved. This can be done only through extensive sensitivity analysis, although was not part of the scope of this research article.

The developed modelling framework can be utilised to explore and quantify the proportion of climate signal to internal system variability. The presence of transients (short-term) and climatic (long-term) trends in the modelled system can contribute to the observed variability in a variety of ways. For example, decreased salinity in the lagoon can result from increased river inflow, precipitation, and periodic salt-water intrusions from the Baltic Sea, which can be identified as amplification of processes, and/or superimposition of short-term and long-term gradual changes at various spatial scales over time. Depending on the driving process, different mitigation and/or adaptation strategies can be employed.

To properly identify and quantify such variability we need to analyse climatic projections and model responses individually, considering temporal and spatial disaggregation. A follow-up study to this will explore this topic fully, where we will utilize statistical techniques, such as the Mann-Kendall test, wavelet analysis, time series decomposition, etc., to identify and quantify trends in the model responses. We will further identify which regional climate model and RCP scenario predicts the most significant changes in the coupled basin-lagoon-sea system. By distinguishing short-term fluctuations from long-term trends, we will be able to evaluate the significance of each factor driving the observed variability. This will help to identify the driving processes behind the changes and, if possible, suggest appropriate mitigation measures.

## 5 Conclusions

The setup of the complex Nemunas River–Curonian Lagoon–Baltic Sea model has been successfully used in the climate change studies presented in this paper. The high-resolution hydrological model for the Nemunas River and hydrodynamic model for the Curonian Lagoon and the Baltic Sea were used. The finite element approach has allowed us to achieve a good compromise of needed resolution and computer efficiency to carry out all needed hydrodynamic simulations. With the help of downscaled climate models, which have been ultimately bias-corrected with observations, the tendency of changes was inferred.

The study results showed that the Curonian Lagoon will be subject to higher Nemunas discharges that will in turn increase the outgoing fluxes into the Baltic Sea. Through these higher fluxes both water residence times and saltwater intrusion events will decrease. Most of these changes will however be only noticeable in the northern part of the lagoon, which are more likely to be influenced by the Nemunas discharge. The southern part of the lagoon will experience much less changes.

The foreseen changes in physical characteristics are not of the scale suggesting significant shifts in the ecosystem structure (dominating species) and functioning (food webs), but expected rather to manifest in some quantitative alterations in the nutrient and organic matter retention capacity. However, some ecosystem services such as ice fishing are expected to be completely vanishing due to the physical constraints.

The study has a limitation as it only considers climatic changes and overlooks socio-economic and basin evolution impacts. However, it still provides a valuable reference point for understanding projected changes in the region. Uncertainty analysis was not within the study's scope but can be addressed through sensitivity analysis. The developed modelling framework allows exploration and quantification of climate signal and internal system variability. Analysing climatic projections and model responses individually, considering temporal and spatial disaggregation, will help identify driving factors for the observed variability. A follow-up study will employ statistical techniques to identify trends and assess significant regional climate models and RCP scenarios. This will guide the identification of driving processes and suggest appropriate mitigation measures.

### Data availability

All numerical modelling results created during this study are openly available in the Zenodo open data repository (https://doi.org/10.5281/zenodo.7500744) as cited in Idzelytė et al. (2023).

### Author contribution

GU and NC initiated the conceptualization and funding acquisition of the research project. RI, NC, and JM performed the analysis and drafted the paper. RI curated the visualization of the results. RI, NC, JM, TD, ARB, AE prepared the original manuscript draft with the assistance of GU. All co-authors reviewed the paper and contributed to the scientific interpretation and discussion.

### Competing interests.

The authors declare that they have no conflict of interest.

### Acknowledgements

This project has received funding from the Research Council of Lithuania (LMTLT), agreement No S-MIP-21-24.

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
