# Peer review of "Coupled hydrological and hydrodynamic modelling application for climate change impact assessment in the Nemunas River watershed— Curonian Lagoon—south-eastern Baltic Sea continuum"

_EGUsphere, 2023_

## Author Response (AR1)

**Referee #1**

Review of

"Coupled hydrological and hydrodynamic modelling application for climate change impact assessment in the Nemunas River watershed– Curonian Lagoon–south-eastern Baltic Sea continuum"

by Idzelyte and co-authors,

The study investigates the effects of climate change on discharge, water temperature, and salinity in the Coronian lagoon as part of the Baltic Sea. They use a new developed high resolution hydrodynamic finite element model setup for the lagoon and part of the southeastern open Baltic Sea. This was coupled to a hydrological discharge model set up for the Nemunas river drainage basin. They demonstrate a freshening of the lagoon due to lower future salt intrusions from the open Sea and increased river discharge which controls the outflow out of the lagoon. The magnitude of these changes dependent on the chosen climate scenario.

The results provide new and important information about the local effects of climate change in the study area which may be of interest also for other marginal lagoons. However, there are a number of shortcomings in the presentation of results, which makes it somewhat difficult for the reader to follow and that should be improved before publication.

*Authors' response:* We are very thankful and highly appreciate the time and effort you have dedicated to provide a valuable, careful and insightful review of our manuscript. We have revised our manuscript according to the critics and recommendations provided and have been able to incorporate changes to reflect most of the suggestions. We strongly believe that the new version of the manuscript is highly improved and is appropriate for publishing for the scientific community.

To summarize, we extensively revised and restructured the manuscript, additional explanations were provided for a better understanding of the system and applied model. The track changes were used for article revision. Below we present the detailed changes in the manuscript according to the comments. We hope that our revisions have addressed all the concerns and have improved the overall quality of the manuscript.

**General comments**

Actually, the results and the discussion remain somewhat descriptive. A lot of detailed information is given but it is difficult for the reader to set this information into a broader scientific context.

Such a context could be given in the Introduction which in it's present form mainly informs about general issues of climate change in the Baltic. May be it's feasible to introduce also to the role of lagoons in the ecosystem of marginal seas to attract researchers from outside the Baltic Sea / Curonian lagoon. What are the specific problems and research questions in the Curonian lagoon (erosion, retention, ventilation was mentioned somewhere later in the text)? Also may be some information about the choice of the climate scenarios. Are they chosen to be most realistic or to be well suited to study strong warming effects etc? Comparing RCP45 and RCP85 allows to assess the effect of moderate climate mitigation. Information about scenarios can be found in e.g.

Thomson AM, Calvin KV, Smith SJ et al (2011) RCP4.5: a pathway for stabilization of radiative forcing by 2100. Clim Change 109:77. https://doi.org/10.1007/s10584-011-0151-4

Riahi K, Rao S, Krey V et al (2011) RCP 8.5—a scenario of comparatively high greenhouse gas emissions. Clim Change 109:33.
https://doi.org/10.1007/s10584-011-0149-y

*Authors' response:* We appreciate your suggestion to include an introduction to the role of lagoons in the ecosystems of marginal seas, as well as explanation of the chosen climate change scenarios, and have added a few brief paragraphs to the introduction of the manuscript that provides an overview of the existing knowledge and  literature on this topic.

The model description is not sufficient. This makes it difficult for the reader to interpret the results later. The meteorological forcing data for the hydrological model should be listed and it should be clear what are the prognostic output variables. It is only stated that temperature and discharge were extracted. Are there more variables of interest (e.g. evaporation which is mentioned in some places of the results). Actually, mainly the setup of SWAT and it's boundary input data are mentioned.

It would be easier to follow if the model descriptions (section 2.3) would be given before section 2.2 boundary data.

*Authors' response:* Thank you for the comment, we value your feedback and have extensively revised the model description section. The meteorological data is listed in Table 1 and is the same for both models as mentioned in sections describing the Climate projection data ("*To feed our developed modeling system*…"). We further add the clarifying sentence in the section describing the Watershed-scale data ("*The same meteorological forcing data, described in Table 1, was used for the hydrological model, as well as for the hydrodynamic.*"). For additional extensive review, please see the section describing the Hydrological model.

As for placing the model description section before the section about the data, we would like to maintain the conventional structure of scientific articles by placing the "model" section after the "data" section, instead of before it.

As I understand a transient simulation from 1970 to 2099 was carried out. I wonder if time series plots could be shown to display the temporal evolution and the variability of some integrated variables (e.g. discharge data, sea level etc) that also would allow to get information about that ratio of the climate signal to it's internal variability.

*Authors' response:* That is a wonderful suggestion and we have plans to explore the uncertainty issue in a separate follow-up study. We can provide this information as an appendix, but rather restrain as we plan to have a separate in-depth study on that.

Likewise, the authors may consider to show some maps of salinity (or their future minus historical anomalies) could be shown. As stated, water temperature is quite homogenous in the lagoon (probably due to the constraint by meteorological air temperature). But are there notable gradients in salinity which would demonstrate the added value of the high resolution FE model?

*Authors' response:* Although there is a slight decrease in salinity levels under RCP scenarios near the outlet of the lagoon, caused by increased freshwater outflow from the Nemunas River into the sea, the overall salinity levels in the south-eastern Baltic Sea remain relatively uniform. As a result, the figure provided below may not exhibit significant variations that would justify its inclusion in the article.

[Figure]

**Specific comments**

**Abstract**

line 18: salt water intrusion into the lagoon?

*Authors' response:* Yes, in the abstract text, the saltwater intrusions imply the Baltic Sea water flowing into the Curonian Lagoon.

*Authors' changes in manuscript:* We adjusted the text to indicate the saltwater flowing into the lagoon:

*"Through these higher fluxes, both the water residence time and saltwater intrusion **into the lagoon** event frequency will decrease."*

**Introduction**

Please give a few more info about the role of lagoons and/or the Curonian lagoon for ecosystem, freshwater supply, nutrient retention, recreation etc) to make this topic interesting for a broader readership and to demonstrate the significance of this area for economy and ecosystem services

*Authors' response:* Thank you very much for the suggestion to include more information about the role of lagoons in the ecosystem of marginal seas for attracting researchers from outside the study area.

*Authors' changes in manuscript:* The introduction section of the manuscript was updated including two new paragraphs:

*"Coastal lagoons and estuaries, forming a continuum between continental and marine aquatic ecosystems (Pérez-Ruzafa et al., 2011) play an important ecological and socio-economic role (Newton et al., 2018; Camacho-Ibar and Rivera-Monroy, 2014). They support important habitats and rich biodiversity (Newton et al., 2018), are nursery grounds and nutrient sources for nearshore fish communities (Meynecke et al., 2007), and are important areas for various biogeochemical processes (Watson et al.,*

*2020; Maher et al., 2019). Coastal lagoons also provide a set of ecosystem services from the physical use of land-/seascape to cultural or aesthetic value (Inácio et al., 2018).*

*In general, lagoon systems comprise 5.3% of the European coastlines with the south-eastern coast of the Baltic being one of the major lagoon regions in Europe (de Wit, 2012). Due to their geographic location, lagoons become very vulnerable to the climate change-related key drivers and arising therefrom physical, ecological, and associated societal disturbances, such as changes in flushing regime, freshwater inputs, water chemistry or loss in communities (Anthony et al., 2009)."*

Pérez-Ruzafa, A., Marcos, C., Pérez-Ruzafa, I.M., Pérez-Marcos, M., 2011. Coastal lagoons: "transitional ecosystems" between transitional and coastal waters. Journal of Coastal Conservation 15, 369–392.

Newton, A., Brito, A., Icely, J., Derolez, V., Clara, I., Angus, S., Schernewski, G., Inácio, M., Lillebø, A.I., Sousa, A., Béjaoui, B., Solidoro, C., Tosic, M., Cañedo-Argüelles, M., Yamamuro, M., Reizopoulou, S., Tseng, H.-C., Canu, D., Roselli, L., Khokhlov, V., 2018. Assessing, quantifying and valuing the ecosystem services of coastal lagoons. Journal for Nature Conservation 44. https://doi.org/10.1016/j.jnc.2018.02.009

Camacho-Ibar, V.F., Rivera-Monroy, V.H., 2014. Coastal Lagoons and Estuaries in Mexico: Processes and Vulnerability. Estuaries and Coasts 37, 1313–1318. https://doi.org/10.1007/s12237-014-9896-0

Meynecke, J.-O., Lee, S.Y., Duke, N.C., Warnken, J., 2007. Relationships between estuarine habitats and coastal fisheries in Queensland, Australia. BULLETIN OF MARINE SCIENCE 80.

Watson, E.B., Hinojosa-Corona, A., Krause, J.R., Herguera, J.C., McDonnell, J., Villegas Manríquez, K.R., Gannon, M.E., Gray, A.B., 2020. Lagoon Biogeochemical Processing is Reflected in Spatial Patterns of Sediment Stable Isotopic Ratios. Journal of Marine Science and Engineering 8, 874. https://doi.org/10.3390/jmse8110874

Maher, D.T., Call, M., Macklin, P., Webb, J.R., Santos, I.R., 2019. Hydrological Versus Biological Drivers of Nutrient and Carbon Dioxide Dynamics in a Coastal Lagoon. Estuaries and Coasts 42, 1015–1031. https://doi.org/10.1007/s12237-019-00532-2

Inácio, M., Schernewski, G., Nazemtseva, Y., Baltranaitė, E., Friedland, R., Benz, J., 2018. Ecosystem services provision today and in the past: a comparative study in two Baltic lagoons. Ecological Research 33, 1255–1274. https://doi.org/10.1007/s11284-018-1643-8

de Wit, R., Mazouni, N., Viaroli, P., 2012. Preface: Research and Management for the Conservation of Coastal Lagoon Ecosystems, South–North Comparisons. Hydrobiologia 699, 1–4. https://doi.org/10.1007/s10750-012-1158-1

Anthony, A., Atwood, J., August, P.V., Byron, C., Cobb, J.S., Foster, C., Fry, C., Gold, A., Hagos, K., Heffner, L., Kellogg, D.Q., Lellis-Dibble, K., Opaluch, J.J., Oviatt, C.A., Pfeiffer-Herbert, A., n.d. Coastal Lagoons and Climate Change: Ecological and Social Ramifications in the U.S. Atlantic and Gulf Coast Ecosystems. Ecology and Society.

*line 33: „Additional climate change impacts…", What is meant with additional. Please give some examples.*

*Authors' response:* Thank you very much for the suggestion to provide more details.

*Authors' changes in manuscript:* The introduction section of the manuscript was updated including some examples of the climate change impacts:

*"For example, climate change-induced increase in precipitation will result in an increased nutrient leaching from agricultural soils, forests, and other land uses (Pihlainen et al., 2020) making it even more difficult to combat eutrophication – a prevalent and serious environmental problem in the Baltic Sea, affecting both the coastal waters and the open sea (McCrackin et al., 2018; Vigouroux et al., 2012)."*

Pihlainen, S., Zandersen, M., Hyytiäinen, K., Andersen, H.E., Bartosova, A., Gustafsson, B., Jabloun, M., McCrackin, M., Meier, H.E.M., Olesen, J.E., Saraiva, S., Swaney, D., Thodsen, H., 2020. Impacts of changing society and climate on nutrient loading to the Baltic Sea. Sci Total Environ 731, 138935. https://doi.org/10.1016/j.scitotenv.2020.138935

McCrackin, M.L., Gustafsson, B.G., Hong, B., Howarth, R.W., Humborg, C., Savchuk, O.P., Svanbäck, A., Swaney, D.P., 2018. Opportunities to reduce nutrient inputs to the Baltic Sea by improving manure use efficiency in agriculture. Reg Environ Change 18, 1843–1854. https://doi.org/10.1007/s10113-018-1308-8

Vigouroux, G., Kari, E., Beltrán-Abaunza, J.M., Uotila, P., Yuan, D., Destouni, G., 2021. Trend correlations for coastal eutrophication and its main local and whole-sea drivers – Application to the Baltic Sea. Science of The Total Environment 779, 146367. https://doi.org/10.1016/j.scitotenv.2021.146367

I also recommend to more introduce to the problems of coupling hydrology and hydrodynamic models in climate applications, and demonstrate why a coupled hydrological model is important in this regard, see e.g.

Hagemann S, Stacke T and Ho-Hagemann HTM (2020) High Resolution Discharge Simulations Over Europe and the Baltic Sea Catchment. Front. Earth Sci. 8:12. doi: 10.3389/feart.2020.00012

*Authors' response:* Thank you for the recommendation, we agree that some information on model coupling should be presented.

*Authors' changes in manuscript:* We adjusted the the text in the Introduction section, discussing more on this matter:

*"The coupling of different numerical models allows researchers to replicate the environmental processes for better assessment of the physical parameters' response to changing climate. **Regarding the climate applications, the inclusion of different models, e.g. hydrological and hydrodynamic, are particularly important since it provides necessary feedback between the atmosphere, land, and ocean, reducing the biases of such estimations. Notably, coupling of a hydrological model into the large modeling systems provides connectivity between the hydrology of the land surface and ocean or sea, making it a crucial component of the coupled system (Hagemann et al., 2020)."***

**Material and methods:**

Please consider section 2.1 to move to the Introductions.

*Authors' response:* The section "2.1 Study area" mostly describes the study domains (Nemunas River watershed, Curonian Lagoon, and south-eastern part of the Baltic Sea) in a generalized manner, thus we believe that this material merits its own section. However, we added some information introducing the reader to the general role of lagoons in the ecosystem of the marginal seas, as well mentioning the specific problems and research questions in the study area.

**Section 2.2.1. Climate projection data.**

Please indicate somewhere that the scenarios you use are from CMIP5 suite of global projections to avoid any confusion with CMIP6 models. Something like:

"The projections are derived according to two Representative Concentration Pathway (RCP) scenarios: RCP4.5 and RCP8.5 of the Coupled model Intercomparison project phase 5.

*Authors' response:* Thank you for the suggestion to clarify the suite of projections.

*Authors' changes in manuscript:* The text in this section was update accordingly:

*"The projections are derived according to two Representative Concentration Pathway (RCP) scenarios: RCP4.5 and RCP8.5 **of the Coupled Model Intercomparison Project - Phase 5 (CMIP5)**."*

line 107. Please include Dieterich et al., 2019

Dieterich, C.; Wang, S.; Schimanke, S.; Gröger, M.; Klein, B.; Hordoir, R.; Samuelsson, P.; Liu, Y.; Axell, L.; Höglund, A.; Meier, H.E.M. Surface Heat Budget over the North Sea in Climate Change Simulations. Atmosphere **2019**, 10, 272. https://doi.org/10.3390/atmos10050272

*Authors' response:* Thank you for the comment, the suggested reference was added to the specified location in the section describing the boundary data.

**2.3.1 Hydrological model**

The description of the hydrological model is very sparse. What are the prognostic variables the model simulates. Which processes are included?

*Authors' response:* Thank you for the suggestion. We have expanded the tool description and modeled processes, while additionally including references that offer an in-depth explanation of the utilized model. The section that describes the hydrological model has undergone significant expansion. We kindly refer you to the updated section in the manuscript for more information.

Please indicate which variables were used as forcing for the hydrological models to simulate the discharge to the lagoon and where these data are derived from. Which are the prognostic variables (if there are more than discharge to the lagoon)?

*Authors' response:* We have extended the section describing the hydrological model and included more details of the tool and modeled processes. We also provide references where the used model is described in depth, as well as data sources used in the setup. Please see the updated section of the hydrological model and the references within.

Please explain what is a "Hydrological Response Unit" of which your setup contains so many. And how is the discharge from SWAT implemented in SHYFEM (is it on one grid cell or distributed over more in accordance to the two nearest basins of the horological model).

*Authors' response:* Thank you for the observation and suggestion. HRUs (Hydrologic Response Units) represent a homogenous area of land in terms of soil type, land use, slope, and administrative unit, with each unit having a unique combination of factors and corresponding hydrologic response parameters that simulate spatially distributed hydrologic processes and water quality, ultimately enhancing the model's ability to predict watershed responses to varying land use and management scenarios. The number of HRUs is great due to the size of the basin (~100 000 sq.km.), the high-resolution of setup data, and the differences between the landscapes, land and water management, and countries.

The discharge from the SWAT model is fed into SHYFEM as an input at the boundary grid points of the SHYFEM model. The output data were used as a direct input for Minija and Nemunas river boundaries, while for other two main rivers discharging in the southern part of the lagoon (Matrosovka and Deima), which are not part of the Nemunas River watershed model, the SWAT output data of the Nemunas River were scaled by the ratio between Matrosovka-Nemunas and Deima-Šešupė calculated from the data presented in Jakimavičius (2012).

Jakimavičius, D.: Changes of water balance elements of the Curonian Lagoon and their forecast due to anthropogenic and natural factors, Kaunas University of Technology, 2012.

*Authors' changes in manuscript:* We included the extensive description of the Hydrological Response Units to the manuscript:

*"**SWAT sub-divides the modeled area into sub-regions, called sub-basins, which are further subdivided into Hydrological Response Units (HRU). HRUs are the basic spatial units used in SWAT to represent a homogeneous area of land with respect to soil type, land use, slope, and administrative unit. Each HRU is characterized by a unique combination of these factors, which influence the hydrological and water quality processes within that unit. Moreover, each HRU has a unique set of hydrologic response**"*

*parameters that govern the water balance within that unit. This approach allows for the simulation of spatially distributed hydrologic processes and water quality, making the model more effective in predicting watershed responses to different land use and management scenarios.*"

**2.3.3 Coupling of models**

It is hard to figure out the coupling from the given information. Could you indicate what variables/integrated parameters are exchanged between SWAT and SHYFEM? SWAT delivers discharge I guess?

*Authors' response:* Yes, you are correct. The water temperature and discharge data from the hydrological model SWAT was used as a direct input for the two rivers (Nemunas and Minija) in the delta area represented as boundaries in the hydrodynamic model SHYFEM. However, the other two main rivers (Matrosovka, branch of the Nemunas, and Deima) discharging in the southern part of the lagoon are not part of the SWAT model, thus the SWAT output data of the Nemunas River were scaled by the ratio between Matrosovka-Nemunas and Deima-Šešupė calculated from the data presented in:

Jakimavičius, D.: Changes of water balance elements of the Curonian Lagoon and their forecast due to anthropogenic and natural factors, Kaunas University of Technology, 2012.

*Authors' changes in manuscript:* The text has been revised to more clearly illustrate the coupling of the hydrological and hydrodynamic models:

"***The hydrological (SWAT) and hydrodynamic (SHYFEM) models were coupled by taking** the daily water temperature and discharge **output** data from the SWAT model **and** directly **using it as input for the** Minija and Nemunas river boundaries **in SHYFEM** (Fig. 3).*"

Also it's not clear how these models can overlap. SHYFEM is a hydrodynamic model covering the Baltic Sea and the lagoon. Is the lagoon also included in the hydrological model? Is it updated by SHYFEM and evaporation is calculated in SWAT?

*Authors' response:* Your observations are astute; there are no overlapping parts between SHYFEM and SWAT. Our intent is to point out that river branches in the Nemunas Delta area are directly simulated by the hydrodynamic model, as the process-based hydrological model is not suitable to resolve the hydrodynamic processes in the delta region.

*Authors' changes in manuscript:* The manuscript text in this section was updated to better indicate the use of hydrodynamic model for simulations in the Nemunas Delta area:

"*A better representation of hydrodynamic conditions in the Nemunas Delta area is achieved with a hydrodynamic model **that simulates the river branches in the delta.**"*

**3.1 Bias correction**

line 185: "...although here we observe that the climate models have their own internal dynamics, thus the correlation with measurements is weak (~0.25). "

please consider removing this statement.

Climate models produce their own variability on synoptical timescales (days to week) for both air temperature and precipitation (as well as for all other prognostic variables). In case of daily temperature

the day to day variability is usually much lower than the seasonal amplitude. Therefore, correlation for temperature is expected to be high, which mainly demonstrates the climate model simulates low temperatures in winter and high temperatures in summer.

In case of precipitation the day to day variations are much higher than the seasonal amplitude. Therefore I guess, if you would have taken daily sums of precipitation you would have get no correlation at all. By using monthly sums of precipitation you smooth out the synoptical variability which "extracts" the seasonal cycle and consequently you get a somewhat higher correlation.

*Authors' response:* Thank you very much for this comment, we fully agree with your opinion and explanation on this matter. We agree that this statement could be misleading to the reader.

*Authors' changes in manuscript:* The suggested statement was removed from the manuscript.

line 194/195: "The differences between the scenarios in the short-term are smaller than in the long-term by an average of 49% under RCP4.5 and ~123% under RCP8.5. " – please rephrase. What does average mean? average over the four climate models?

*Authors' response:* It means that the percentage differences of each month were averaged.

*Authors' changes in manuscript:* The text was updated to clarify the averaging of monthly percentage difference:

*"The differences between the scenarios in the short-term are smaller than in the long-term by **a monthly average of 49% under RCP4.5 and ~123% under RCP8.5.**"*

line 201: Please add a short sentence highlighting in what the context of this information is important. What is the role of land - sea difference?

*Authors' response:* In general, the land-sea difference refers to the contrast in physical characteristics and processes between land and water bodies, playing a critical role in shaping many aspects of the Earth's climate, weather patterns, and ecosystem dynamics. Some of the key roles of the land-sea difference include: i) climate regulation – the ocean's large heat capacity and its ability to circulate heat through ocean currents help to regulate global temperatures, which in turn affects weather patterns; ii) precipitation patterns – the contrast in heating and cooling rates between land and sea can cause differences in air pressure, leading to the formation of weather systems and precipitation patterns.

*Authors' changes in manuscript:* We adjusted the the text, shortly presenting the role of land-sea difference:

*"**The difference between land and sea regulate local climate (temperature) and precipitation patterns.** Considering the differences between marine and terrestrial parts of the study area, there is a clear pattern of lower air temperatures (on average by ~1–1.5 °C) over the Baltic Sea and the Curonian Lagoon…"*

Table 2 contains a lot of numbers but is discussed only very briefly. Perhaps a figure instead of a table would provide more info about changes in seasonallity etc. Also, yearly changes could be pointed out as this could be of interest for the long term changes in the freshwater inventory.

*Authors' response:* The table displays the characteristics or elements that you have requested to be included in the graphical representation. However, as we aim to provide a more detailed account of the temporal evolution of these parameters in a subsequent article, we would prefer to abstain from adding

any additional figure to this manuscript. However, subject to the editor's approval, we could enhance the table by utilizing a heatmap to add colors to the cells.

**Air temperature (°C)**

| | | Jan | Feb | Mar | Apr | May | Jun | Jul | Aug | Sep | Oct | Nov | Dec | Year |
|---|---|---|---|---|---|---|---|---|---|---|---|---|---|---|
| Historical (1975-2005) | | -2.6 | -2.6 | 0.3 | 5.3 | 10.7 | 14.6 | 17.9 | 17.5 | 13.2 | 8.1 | 1.8 | -2.3 | 6.8 |
| Short-term | RCP4.5 | -0.5 | 0.4 | 2.4 | 7.4 | 12.2 | 15.9 | 19.2 | 18.7 | 14.6 | 9.7 | 3.8 | 0.0 | 8.6 |
| (2020-2050) | RCP8.5 | -0.6 | 0.5 | 2.8 | 7.5 | 12.4 | 16.1 | 19.2 | 18.8 | 14.6 | 10.0 | 4.2 | -0.2 | 8.8 |
| Long-term | RCP4.5 | 0.9 | 1.4 | 3.4 | 8.4 | 13.3 | 16.9 | 20.1 | 19.7 | 15.5 | 10.9 | 4.7 | 1.0 | 9.7 |
| (2070-2100) | RCP8.5 | 2.9 | 3.4 | 5.0 | 10.0 | 14.7 | 18.1 | 21.8 | 21.5 | 17.6 | 12.4 | 6.7 | 3.3 | 11.5 |

**Precipitation (mm month$^{-1}$)**

| | | Jan | Feb | Mar | Apr | May | Jun | Jul | Aug | Sep | Oct | Nov | Dec | Year |
|---|---|---|---|---|---|---|---|---|---|---|---|---|---|---|
| Historical (1975-2005) | | 42.8 | 36.5 | 43.1 | 36.9 | 55.9 | 87.1 | 57.5 | 57.9 | 48.7 | 63.7 | 39.0 | 48.7 | 617.7 |
| Short-term | RCP4.5 | 54.4 | 47.6 | 53.7 | 41.1 | 53.1 | 83.4 | 69.9 | 64.7 | 55.6 | 72.7 | 49.9 | 56.4 | 702.5 |
| (2020-2050) | RCP8.5 | 53.1 | 46.9 | 50.1 | 43.1 | 61.0 | 83.1 | 75.4 | 69.1 | 54.2 | 71.2 | 50.5 | 58.4 | 716.2 |
| Long-term | RCP4.5 | 56.9 | 46.5 | 53.7 | 44.8 | 58.8 | 83.3 | 69.2 | 70.0 | 54.0 | 78.2 | 53.6 | 59.3 | 727.3 |
| (2070-2100) | RCP8.5 | 72.3 | 52.2 | 65.4 | 49.5 | 67.7 | 93.2 | 73.6 | 77.9 | 60.6 | 85.3 | 62.7 | 70.3 | 830.8 |

line 214: what is meant by elements?

*Authors' response:* The meteorological elements are what we imply as "elements." A few examples of meteorological elements are temperature, pressure, humidity, wind direction and speed, radiation, and phenomena (such as precipitation, snowstorms, etc.).

*Authors' changes in manuscript:* The text was updated to give an example of the elements:

*"The increased liquid precipitation in winter will likely lead to an increased soil erosion and nutrient wash-off, when the soil is bare and exposed to the elements, **such as rain and wind**."*

line 217ff: You could discuss the general pattern in a broader context. For example, the opposite seasonal trends, i.e. the higher discharge during winter, and the reduced discharge during summer well reflect the changes in the water cycle over eastern Europe for which dryer conditions during summer and wetter conditions during winter are projected with a strong surplus of precipitation in the yearly sum. (e.g. Jacob et al., 2014, Gröger et al., 2021, Christensen, 2022).

Jacob, D., Petersen, J., Eggert, B., Alias, A., Christensen, O. B., Bouwer, L. M., Braun, A., Colette, A., Déqué, M., Georgievski, G., Georgopoulou, E., Gobiet, A., Menut, L., Nikulin, G., Haensler, A., Hempelmann, N., Jones, C., Keuler, K., Kovats, S., Kröner, N., Kotlarski, S., Kriegsmann, A., Martin, E., van Meijgaard, E., Moseley, C., Pfeifer, S., Preuschmann, S., Radermacher, C., Radtke, K., Rechid, D., Rounsevell, M., Samuelsson, P., Somot, S., Soussana, J.-F., Teichmann, C., Valentini, R., Vautard, R., Weber, B., and Yiou, P.: EURO-CORDEX: new high-resolution climate change projections for European impact research, Reg. Environ. Change, 14, 563–578, https://doi.org/10.1007/s10113-013-0499-2, 2014
Gröger, M., Dieterich, C. & Meier, H.E.M. Is interactive air sea coupling relevant for simulating the future climate of Europe?. *Clim Dyn* **56**, 491–514 (2021). https://doi.org/10.1007/s00382-020-05489-8
Christensen, O. B., Kjellström, E., Dieterich, C., Gröger, M., and Meier, H. E. M.: Atmospheric regional climate projections for the Baltic Sea region until 2100, Earth Syst. Dynam., 13, 133–157, https://doi.org/10.5194/esd-13-133-2022, 2022.

*Authors' response:* Thank you for your valuable suggestion to discuss the general pattern in a broader context. We have incorporated this discussion into our manuscript. By providing additional context to the observed patterns of precipitation, we hope to enhance the readers' understanding of the potential implications for the future. Please refer to the extensively revised section 4.1 Local climate insights.

line 226 and previous occurrences: what is meant by study domains? Does it refer to SWAT hydrological model and the hydrological SHYFEM models?

*Authors' response:* Study domains refer to the Nemunas River watershed, Curonian Lagoon, and south-eastern part of the Baltic Sea.

*Authors' changes in manuscript:* We included a notice in the beginning of section "2.1 Study area":

*"The study site covers **three domains** – part of the south-eastern Baltic Sea region together with the Curonian Lagoon and the Nemunas River watershed (Fig. 1)."*

**Figs 6 & 7:** I am not sure how to interpret these figures. Please denote what is the reference volume from where water flows in or out. Considering section 4 (LT-RU-border), and provided SHYFEM conserves the water volume, the net water exchange across this section would be close to zero. But Figure 6 shows a deficit, since there is much more outflow than inflow. So where is this deficit balanced? Is it the riverine water input from the hydrological discharge model?

*Authors' response:* Figure 7 displays the percentage change of water flux for the scenario runs, both short-term and long-term, as compared to the historical period. We acknowledge that Figure 6 might be difficult to understand for the reader, but the general idea is to show the percentage of inflowing and outflowing flux of the absolute flux, that outflowing water is dominating in the lagoon. There is a deficit for the LT-RU border, but it is not much more outflow than inflow. This (small) deficit is due to the fact that there are also some rivers entering the Russian area (Deima and a southern branch of the Nemunas - Matrosovka). Therefore it is not surprising that this imbalance exists. However, it is much smaller than the imbalance in the other areas, where the much higher Nemunas inflow leads to higher outflow than inflow.

*Authors' changes in manuscript:* Figure 6 was updated showing the average inflow and outflow flux (in m³ s⁻¹) during the analysis periods, including its percentage of the absolute flux.

[Figure]

*Figure 6. Average outflowing and inflowing water volume (hatched pattern) including its percentage of the absolute flux (solid pattern) of each analysis period – historical (1975–2005), short-term (2020–2050), and long-term (2070–2100) under RCP4.5 and RCP8.5 scenarios. The locations of cross-sections are shown in Fig. 7.*

**Table 3:** Please give the some explanation how to interpret the results. Do out and inflow have to be balanced?

*Authors' response:* In the long run, it is imperative to maintain a balance between inflow and outflow, taking into account factors such as river discharge and evaporation/precipitation. By incorporating the Nemunas River discharge, it can be demonstrated that the inflow and outflow are indeed not balanced.

**Section 3.4 Salinity**

The results for salinity should be discussed in the context of uncertainty. As I understood, lateral open sea boundaries for were prescribed from results of Gröger et al. 2019. These scenarios do not contain the effects of global sea level rise. This should be discussed. See literature for example Meier et al. 2021. In short: rising global see levels would lead to higher salinities in the Baltic Sea due to higher volume inflow from the North Sea.

Markus Meier, H.E., Dieterich, C. & Gröger, M. Natural variability is a large source of uncertainty in future projections of hypoxia in the Baltic Sea. Commun Earth Environ **2**, 50 (2021). https://doi.org/10.1038/s43247-021-00115-9

*Authors' response:* Thank you very much for the suggestion and recommendation.

*Authors' changes in manuscript:* We included this consideration in the discussion section 4.5 Salinity dynamics:

*"As seen and suggested from other Baltic Sea salinity projection studies, the salinity in the Baltic does not have a distinct pattern of changing in the future and the projections have a lot of uncertainties (Lehmann et al., 2022; Meier et al., 2022b). Our study of salinity changes in the south-eastern Baltic Sea demonstrates a decreasing tendency, especially in the coastal area, northward from the lagoon outlet, where the sea water is diluted by discharging freshwater from the lagoon (Table 8).* **However, the scenarios that we used to run the hydrodynamic model did not consider the global sea level rise, which according to Meier et al. (2021) could more or less outweigh the projected increase of the river runoff into the Baltic Sea, due to the higher volume of inflow from the adjacent North Sea.**"

line 269: Please rephrase the sentence. When there are no water intrusions from the Baltic Sea to the lagoon, how can salt intrusions be calculated?

*Authors' response:* Thank you very much for the observation, indeed the explanation might be misleading. By saltwater intrusions we meant the salinity values above 2 g kg$^{-1}$.

*Authors' changes in manuscript:* The text was changed as follows:

*"Seasonally, the highest change of saltwater intrusion days is observed during winter and especially spring, when the outflow of the lagoon is dominating and the* **salinity values in Juodkrantė do not exceed 2 g kg$^{-1}$** *(right side of Fig. 8)."*

**Section 3.5 Water temperatures**

Please consider to give changes in temperature units instead of relative % increases. It is not surprising that the temperature is homogenous as the water surface is tightly constraint by the atmospheric temperature (in particular when the lagoon water body is small). But how does this look like for salinity? SHYFEM has extremely high resolution which is a prerequisite to represent steep salinity gradients.

*Authors' response:* Thank you for the suggestion to present water temperature results in temperature units, we incorporated your recommendations in the manuscript.

High-resolution models such as SHYFEM can better represent steep salinity gradients and capture the small-scale features that affect the salinity distribution in a water body. Salinity gradients are only existing in the north of the Curonian Lagoon, close to the Klaipėda Strait. Due to the freshwater discharge of the Nemunas river, both the southern lagoon and the Nemunas Delta in the south have a salinity level that is almost negligible (close to 0 g kg⁻¹). In the Klaipeda Strait, however, the gradients between the Baltic Sea salinity (around 6-8 g kg⁻¹) and the one in the Curonian Lagoon (close to 0 g kg⁻¹) can form. This happens, however, in a very narrow area, inside the strait. As mentioned in the article, salinity intrusions happen at some times of the year, but they are also washed out very fast due to the freshwater influx. Therefore, in summary, also salinity gradients are very localized and normally the lagoon shows a quite homogeneous picture of spatial salinity.

*Authors' changes in manuscript:* The water temperature results section was adapted to indicate the degree Celsius change:

*"Considering the seasonal dynamics, water temperature is projected to increase the most during winter, by ~60% in the short-term and more than twice in the long-term (Table 5). During spring and autumn, the increase is lower – ~17% in the short-term and 19–27% in the long-term. The lowest increase is projected during summer season – 5–6% in the short-term and 9–13% in the long-term.* **However, considering the water temperature increase in °C, we can see that it in the short-term it is similar between the seasons ranging from on average 0.9 °C (in winter) to 1.4 °C (in spring), while in the long-term the change is higher – 1.9 °C in winter and spring, 2.2 °C in summer, and .3°C in autumn.** *The more apparent difference between the RCP scenarios is observed during the long-term, where RCP8.5 displays a higher temperature increase."*

**section 3.6 Water level**

line 313: what is the reference level and where do these values come from? From the model, literature, observations? Are extreme values included like e.g. during storms?

*Authors' response:* Since we used boundary data from a high resolution regional coupled ocean–sea ice–atmosphere model RCA4–NEMO (Gröger et al., 2019), the reference level is the one from the climatic simulations. This level agrees quite well with the actual observed value in the Baltic Sea. Extreme values are also included. Clearly, the extreme values modeled by our model depend crucially on the simulated climate runs used for the boundary conditions. The values shown in the text are the hydrodynamic model results.

Gröger, M., Arneborg, L., Dieterich, C., Höglund, A., and Meier, H. E. M.: Summer hydrographic changes in the Baltic Sea, Kattegat and Skagerrak projected in an ensemble of climate scenarios downscaled with a coupled regional ocean–sea ice–atmosphere model, Clim. Dyn., 53, 5945–5966, https://doi.org/10.1007/s00382-019-04908-9, 2019.

**Table 6:** again what is the reference level. Is it perhaps the border between the lagoon and the open sea?

*Authors' response:* Again, as described above, the reference level is of the climatic simulations from the high resolution regional coupled ocean–sea ice–atmosphere model RCA4–NEMO (Gröger et al., 2019).

What is the idea of averaging 5 stations in the Curonian together with area average of the SE baltic Sea? How can this be interpreted?

*Authors' response:* The results in the Baltic Sea were averaged over five pre-selected discrete locations distributed throughout the sea. This approach was taken to ensure a representative sample of the region of interest. The resulting averages provide a reliable estimate of the overall conditions in the area. The decision to present the point locations in the Curonian Lagoon separately was made in order to provide a more precise and detailed depiction of the area. However, since the point statistics in the lagoon are very similar, we updated table 6 showing the average water level in the lagoon.

*Authors' changes in manuscript:* Table 6 and its caption was updated as follows:

| | 1975-2005 | 2020-2050 | | 2070-2100 | |
|---|---|---|---|---|---|
| | Historical | RCP4.5 | RCP8.5 | RCP4.5 | RCP8.5 |
| Baltic Sea | -0.15 | 0.01 | 0.03 | 0.17 | 0.23 |
| Klaipėda Strait | -0.13 | 0.04 | 0.06 | 0.19 | 0.26 |
| Curonian Lagoon | -0.07 | 0.10 | 0.13 | 0.25 | 0.34 |

*Table 6. The average water level during the historical (1975–2005) and the short- (2020–2050) and long-term (2070–2100) periods under RCP4.5 and RCP8.5 scenarios. **Data were averaged over the south-eastern Baltic Sea, Klaipėda Strait, and the Curonian Lagoon.***

**Section 3.7 Water residence time**

How were these data derived? Usually I would assume you need a total volume of the water body and water fluxes into/out of it. Are the values in Table 7 for the entire SHYFEM domain, only for the lagoon or for the open SE Baltic Sea?

*Authors' response:* In simple terms, a concentration is released in the basin and the decay of the concentration due to water fluxes is computed. By fitting the decay curve to an exponential curve, the time scale of the exponential function can be computed, which represents the residence time. The methodology is explained in Cucco and Umgiesser, 2006. The values in Table 7 are not for the entire SHYFEM domain, but only for the northern and southern parts of the lagoon, as well as for the whole lagoon. The open south-eastern Baltic Sea was not included in these computations.

Cucco, A. and Umgiesser, G.: Modeling the Venice Lagoon residence time, Ecological Modelling, 193, 34–51, https://doi.org/10.1016/j.ecolmodel.2005.07.043

*Authors' changes in manuscript:* We updated the beginning of this section, indicating that the analysis is done for the Curonian Lagoon:

*"The water residence time (WRT) **in the Curonian Lagoon** generally is likely to have a decreasing tendency..."*

**4 Discussion**

line 364: I am puzzled here. Before I got the impression the results were derived with the coupled SHYFEM – SWAT model which used input data from the downscaled global models.

*Authors' response:* Indeed you are correct, the results of this study were derived using a coupled hydrological-hydrodynamic (SWAT-SHYFEM) modeling system, which was run using climate data from regional climate models. This section of the discussion focuses on the biases between the output of the regional climate models and local observations, however, we agree that it could be confusing for the reader.

*Authors' changes in manuscript:* We updated the text by clarifying the context of this discussion section:

*"The results of this study have been achieved **by applying a coupled hydrological-hydrodynamic modeling system, which was run** with **input data from** regional climate models that have been downscaled from global models. The quality of the **regional climate** model output was generally good, but…"*

**4.2 Water flow**

line 388. It is clear that outflow of the lagoon dominates when so much river discharge takes place. This is also consistent with the reduced water residence times. Therefore you may consider merge section 4.2 with 4.6

*Authors' response:* Thank you for your valuable feedback on our article. We appreciate your suggestion to merge the water flow and water residence time sections in the Discussion part of the article. However, we would like to keep these sections separate as the water flow section is already quite extensive. To address your concern, we propose to move the water residence time section after the water flow section in the Discussion as well as Results section of the manuscript. This way, we can maintain the organization and coherence of the article while still addressing the important topic of water residence time.

line 405. The higher risk for flooding in winter is a nice result which could be highlighted in the abstract as well.

*Authors' response:* Thank you for the comment and recommendation.

*Authors' changes in manuscript:* We included a sentence in the abstract highlighting the increased risk of flooding in the Nemunas delta area:

*"Most of these changes will be more pronounced in the northern part of the lagoon, which is more likely to be influenced by the variations in the Nemunas River discharge. **Its delta area may be susceptible to flooding as a result of the elevated discharge during winter.**"*

**4.3 water level dynamics**

last sentence: why leads a fixed boundary to the Baltic Sea to an overestimation of the increase in sea level.

*Authors' response:* Assuming a fixed boundary disregards the possibility of sea level rise and coastal erosion, leading to overestimation of sea level increase. This happens as the coastline cannot adapt to rising water levels by moving inland. Considering the potential for coastal erosion and landward shift is crucial for accurate sea level rise projections, without this it may lead to overestimation of the impact on coastal areas.

**4.4 Salinity dynamics**

Salinity in the open SE Baltic is much controlled by the lateral boundary data which do not include the effect of global mean sea level rise. Therefore the salinity drops in the future is likely overestimated. Please

discuss the uncertainties for salinity in the light of lacking effect of global mean sea level rise (Meier et al., 2021).

*Authors' response:* Thank you very much for the suggestion and recommendation. We included this information in the discussion about salinity projections.

*Authors' changes in manuscript:* We updated the discussion as follows:

"*As seen and suggested from other Baltic Sea salinity projection studies, the salinity in the Baltic does not have a distinct pattern of changing in the future and the projections have a lot of uncertainties (Lehmann et al., 2022; Meier et al., 2022b). Our study of salinity changes in the south-eastern Baltic Sea demonstrates a decreasing tendency, especially in the coastal area, northward from the lagoon outlet, where the sea water is diluted by discharging freshwater from the lagoon (Table 8). **However, the scenarios that we used to run the hydrodynamic model did not consider the global sea level rise, which according to Meier et al. (2021) could more or less outweigh the projected increase of the river runoff into the Baltic Sea, due to the higher volume of inflow from the adjacent North Sea.**"*

**4.5 Water temperature dynamics**

This paragraph is mainly descriptive. Please consider to move it to the results section.

*Authors' response:* Thank you for the suggestion, we revised this section and shortened the discussion of our derived results.

*Authors' changes in manuscript:* The section was shortened and summarized as follows:

"*The mean temperature of the water column in the Klaipėda Strait is less sensitive to changes, **while** the lagoon is **expected** to **see an** increase **in temperature** under both RCP scenarios (Table 8). **Long-term projections under RCP8.5 show the highest increases. The south-eastern Baltic Sea also shows increasing temperature patterns, with the highest changes occurring during winter and spring. Coastal areas have similar dynamics, but with higher changes compared with the historical period. Coastal average water temperature is lower by 1.5 °C in winter and higher by 4.9 °C during the rest of the year compared to the south-eastern Baltic Sea.**"*

**4.6 Residence time**

line 448: Isn't the increased outflow ultimately rather a result of the increased discharge from the rivers to the lagoon?

*Authors' response:* Yes, you are correct. We agree that the description could be perceived as misleading to the reader.

*Authors' changes in manuscript:* The sentence in this section was updated representing the continuity of flowing water from the Nemunas River watershed into the Baltic Sea:

"*As in the case with saltwater intrusions, this is due to the increased **discharge** from the **Nemunas River into the lagoon and further outflowing** to the sea, where WRT are less varying in the north, whereas it decreased sensibly in the south.*"

**4.7. Ice thickness**

Indeed, it is very likely that the ice season is affected. Is it possible to analyze the length of the season with ice presence out from the model outputs?

*Authors' response:* Yes, it is possible to analyze the number of days of ice presence in the lagoon from the model output.

*Authors' changes in manuscript:* The changes of ice duration in the Curonian Lagoon were explained in the Discussion section of the manuscript as well as presented in the Results section "3.8 Ice thickness" including the projections of the ice season duration changes in the lagoon:

*"The average maximum ice thickness and **ice season duration** during the short-term period can decrease by 25% compared with the historical period. /.../ **The ice duration is as well projected to continue becoming shorter by 35% under RCP4.5 and 67% under RCP8.5 scenario, compared with its duration during the historical period.** These changes indicate a possible severe modification of ice regime in the lagoon, leading to winter seasons with a more likely unstable thin ice cover."*

**4.8 ecosystem impact**

"The foreseen salinity changes in the mostly freshwater ecosystem are not of the scale suggesting significant shifts in the ecosystem functioning even in the northern part of the lagoon "

This is a strong statement. Please consider to formulate this weaker.

*Authors' response:* Thank you for the suggestion.

*Authors' changes in manuscript:* The sentence has been reformulated accordingly:

*"The foreseen **minor** salinity changes in the mostly freshwater ecosystem are **unlikely to cause** significant shifts in the ecosystem functioning even in the northern part of the lagoon."*

lines 469 to 447: You state that the results of Ivanauskas, 2022 are in contrast to predictions of Baltic Sea fishery. But it's not clear why. And can your results be used to explain this contrast. I do not understand what is the point with this paragraph.

*Authors' response:* The main reason for this contrast is that Curonianian lagoon fishery is dominated by fish species like pikeperch and bream preferring warmer water conditions in contrast to the Baltic sea, especially the north-eastern part. The present structure of fish communities in the Curonian lagoon are less sensitive to the climatic changes.

*Authors' changes in manuscript:* The revised paragraph now more clearly expresses the contrast:

*"A recent analysis (Ivanauskas et al., 2022) indicated a statistical relationship between WRT and salinity and both catches and populations of main commercial fish populations **also stating that predicted decrease in salinity and increasing temperatures are favorable for main commercial fish species.** This is in stark contrast to the predictions for the Baltic Sea fishery, where catches are expected to be decreasing in quantity, and especially in the quality (Climate Change in the Baltic Sea, 2021)."*

lines 475ff. Again, you mention the shortening of the ice cover period. This is exactly what would be expected in warming scenarios and not very surprising. Can you give a more quantitative measure for this, like e.g number of days with sea ice or shifts in the start/end of the ice season for the lagoon etc? You use a model which is well suited for the lagoon and so this would yield an added value to the current knowledge.

*Authors' response:* As mentioned above, the number of days of ice presence in the lagoon were added to the Results section "3.8 Ice thickness". The detailed description of ice phenology (shifts in the start/end, duration of the ice season, projected thickness changes) can be found in previous studies, dedicated solely for the ice cover season dynamics and thickness projections in the Curonian Lagoon:

Idzelytė, R., Kozlov, I. E., and Umgiesser, G.: Remote Sensing of Ice Phenology and Dynamics of Europe's Largest Coastal Lagoon (The Curonian Lagoon), Remote Sens., 11, 2059, https://doi.org/10.3390/rs11172059, 2019.

Idzelytė, R. and Umgiesser, G.: Application of an ice thermodynamic model to a shallow freshwater lagoon, Boreal Environ. Res., 26, 61–77, 2021.

To reach a broader readership it would be good to be a bit more verbose about the speculations on retention and nutrient cycling. What are the processes that determines retention and how do the drivers change according to your results and what can be expected.

*Authors' response:* The processes that determine nutrient retention in water bodies are complex and influenced by a variety of factors. The drivers of nutrient retention change according to the water residence time, and external factors such as land use and climate change can also play a significant role in shaping nutrient cycling in aquatic ecosystems. The retention of both nutrients and organic matter is found to be negatively related to the water residence time, which in turn would lead to the decrease in regulation ecosystem services.

*Authors' changes in manuscript:* We elaborated on this topic more in the manuscript:

*"The latest studies, revealing the mechanisms of nutrient cycling and eutrophication processes (Bartoli et al., 2018), point toward the expected reduction of the role of the Curonian Lagoon as a coastal filter. The shortening of the ice cover period will reduce the system denitrification capacity while the decrease of WRT will lead to substantial reduction of nutrient **and organic matter** retention **due to the decreased burial and absorption rates**. So far, the only difference is the more favorable conditions for the cage aquaculture in the lagoon, where the decreased WRT and increased flushing in combination with modern practices of multi trophic aquaculture (e.g. floating vegetable gardens) could remove some of the constraints preventing the development of cage aquaculture existing at present."*

**5 Conclusions**

line 491: What kinds of ecosystem functioning shifts do you refer here?

*Authors' response:* We refer to the shifts of ecosystem structure and food webs as the ecosystem functioning.

*Authors' changes in manuscript:* We made revisions to the manuscript to convey this idea with greater clarity:

*"The foreseen changes in physical characteristics are not of the scale suggesting significant shifts in the ecosystem **structure (dominating species) and** functioning **(food webs)**, but expected rather to manifest in some quantitative alterations in the nutrient **and organic matter** retention capacity."*

you should also mention here that you did not consider mitigation scenarios, i.e. the RCP2.6 scenario which is in agreement with the goal to limit global mean climate warming to 2.0 degree compared to the preindustrial level. See e.g.

van Vuuren DP, Stehfest E, den Elzen MGJ et al (2011) RCP2.6: exploring the possibility to keep global mean temperature increase below 2 °C. Clim Change 109:95. https://doi.org/10.1007/s10584-011-0152-3

*Authors' response:* Thank you for the remark. Indeed, the mitigation scenario was not mentioned in the manuscript.

*Authors' changes in manuscript:* We updated the conclusions section informing the reader once again that the results were derived considering RCP4,4 and RCP8.5 scenarios, lcsking the RCP2.6:

**"*It is worth mentioning that the analysed physical changes in the presented continuum system were assessed in the scope of climate change under RCP4.5 and RCP8.5 scenarios; the RCP2.6 mitigation scenario (van Vuuren et al., 2011) was not covered in this study. ….*"**

line 495: This sentence could be removed if no information is given what kind of extensive sensitivity analyses are envisaged.

*Authors' response:* Thank you for your suggestion on this matter. We removed the last sentence from the conclusions section of the manuscript.

As I understand, the residence time in the lagoon is mainly controlled by river discharge implying the changes in wind driven circulation are less important (contrary to the open sea). This might be also of interest for other lagoons that may suffer under episodic hypoxic events and for which decreasing discharges are predict under future scenarios (even if this is not a problem in the Curonian lagoon).

*Authors' response:* Yes, the river discharge is the main factor shaping the residence times in the northern part of the Curonian lagoon. In the southern part, which has a much larger area, the wind impact should also be recognised.

**Referee #2**

The ms entitled "Coupled hydrological and hydrodynamic modelling application for climate change impact assessment in the Nemunas River watershed–Curonian Lagoon–south-eastern Baltic Sea continuum" by Idzelyte et al. focuses on the analysis of the effects of climate change on a Baltic coastal region using modelling tools. The approach is particularly interesting because it does not focus only on one water body, as is usually the case, but, using coupled hydrological and hydrodynamic models, covers the land-sea gradient, analysing the expected changes in the catchment, the Nemunas River, the Curonian Lagoon and the adjacent coastal Baltic sea, in an integrated way, considering the interactions between them.

The work is well laid out and the data used to feed and set the boundary conditions of the model are well documented and presented in a detailed way. The two coupled models used, SWAT and SHYPHEM, are extensively tested in their respective applications, the former for catchment and runoff models and the latter in several coastal lagoons, especially in the Mediterranean, but also in the Curonian lagoon.

*Authors' response:* We extend our sincere thanks to you for dedicating your valuable time and effort to review our manuscript. Your expertise and commitment were evident in the thoughtful and constructive feedback you provided, and we are grateful for your guidance and support throughout the process.

Your detailed feedback and suggestions have been instrumental in improving the quality of our work, and we appreciate the valuable insights you shared. Your comments have helped us refine our research and strengthen our arguments, and we have carefully considered each one of them in the revision process.

Thanks to your input, we have been able to broaden our perspective and made revisions to the manuscript accordingly. Once again, we appreciate your efforts and contribution to our work, and we are grateful for the opportunity to benefit from your expertise.

Perhaps, one of the weaknesses of the work would be that the future scenarios consider only climatic conditions, but not the evolution of the drainage basin in accordance with them or socio-economic development. These changes will undoubtedly affect land use, water requirements for different uses and, in general, surface runoff and the functioning of aquifers. Although this involves that the predictions cannot be those that will actually occur, this does not invalidate the work since it gives the frame of reference in which they will occur, and can make the article a necessary reference for more socio-economic predictive models.

Perhaps, these expected changes, or in which direction can they affect, should be discussed in the Discussion chapter from the knowledge of how SWAT model responds to such variables although assuming that it must be just a series of considerations just to take it into account, more than a real analyses if there are no real data, to avoid to be speculative.

*Authors' response:* Thank you for your comment. While considering drainage basin and socio-economic development would be valuable for future scenarios, we focused only on climatic conditions due to the high uncertainty and difficulty in predicting socio-economic development accurately. We aimed to focus on the more measurable and predictable factors of climate change and its potential impact on the region. Nevertheless, we acknowledge the potential impact of these factors on land use, water requirements, surface runoff, and aquifer functioning. We plan to explore these topics in a future article. Thank you for your feedback, and we hope our article serves as a useful starting point for further research and discussion.

*Authors' changes in manuscript:* We amend the article's discussion on this matter:

"***One potential limitation of the study is that the future scenarios only account for changes in climatic conditions, without taking into consideration the evolution of the drainage basin, or the changes in the lagoon and the port area due to socio-economic development. These factors are expected to impact land use, water requirements, surface runoff, and aquifer functioning, as well as hydrodynamic conditions in the lagoon-sea interface. Considering the political situation of the neighboring countries, socio-economic changes in this region are extremely difficult to predict. While this means that the predictions may not precisely reflect what will happen, the study still provides a valuable reference point for understanding the projected changes in the region. It can also serve as a necessary starting point for the development of more comprehensive socio-economic predictive models.***"

One of the aspects included in the discussion is 4.8 Impacts on the ecosystem structure and functions. To consider this is important for having a complete view of the real consequences of the climate change and to design management actions to avoid negative consequences for the biodiversity and human uses as fisheries. Authors assume that it is more complicated to predict but they state that "However, the foreseen changes in physical characteristics are not of the scale suggesting significant shifts in the ecosystem functioning, but are expected to manifest in some quantitative alterations in the nutrient retention capacity".  This statement should perhaps be qualified and discussed.

*Authors' response:* Thank you for bringing up this point. Climate change's impact on ecosystems is crucial to consider, requiring a complete understanding to create effective management strategies that prevent negative outcomes for both biodiversity and human activities, such as fisheries.

*Authors' changes in manuscript:* The wording of this statement was altered in order to improve its clarity:

"*The foreseen changes in physical characteristics are not of the scale suggesting significant shifts in the ecosystem **structure (dominating species) and** functioning **(food webs)**, but expected rather to manifest in some quantitative alterations in the nutrient **and organic matter** retention capacity. However, some*

*ecosystem services such as ice fishing are expected to be completely vanishing due to the physical constraints"*

The modelling results show that the Baltic Sea saltwater intrusions into the Curonian Lagoon will likely decrease in the future and seasonally, the highest change of saltwater intrusion days is observed during winter and especially spring.

The authors address here the consequences on pelagic production and fisheries, and nutrient fluxes in sediments, but do not consider the implications on lagoon-sea connectivity and migratory movements, or larval exchanges between the lagoon and sea. There is already some papers demonstrating the importance of connectivity between lagoons and sea in the structure of ichthyoplanktonic assemblages and genetic population structure of the fauna. Although going into these aspects would require Lagrangian models, a discussion on the implications of changes in residence times might be possible. It is evident that since the results show a decrease in marine intrusions in the future and that the greatest changes materialize during the spring (typical time of larval production), these changes must have consequences on the genetic structure of the populations and on the community structure.

*Authors' response:* Indeed, the connectivity between the Baltic Sea and the Curonian Lagoon, will likely to be altered due to the change in the projected hydrodynamic conditions, i.e. increasing outflow from the lagoon to the sea and decreased saltwater inflow into the lagoon, which is important for maintaining genetic diversity and connectivity between the populations. The degree of this connectivity can have significant implications for migratory movements, larval exchanges, and the overall health and biodiversity of the marine ecosystem. As you mentioned, we have included the reference to the recent paper into the discussion regarding the implications on the connectivity of lagoon-sea fish population.

*Authors' changes in manuscript:* The discussion chapter on Impacts on the ecosystem structure and functions was updated including your suggested topic:

"***According to the latest combined genetic and hydraulic modelling study (Fernández-Alías et al., 2022), the connectivity between the Curonian lagoon and coastal Baltic sea fish population could be altered due to the decrease in marine intrusions during the spring spawning period common to the most cyprinid species common for the Curonian lagoon.***"

Fernández-Alías, A., A. Razinkovas-Baziukas, R. Morkūnė, H. Ibáñez-Martínez, E. Bacevičius, I. Muñoz, C. Marcos, A. Pérez-Ruzafa.: Recolonization origin and reproductive locations, but not isolation from the sea, lead to genetic structure in migratory lagoonal fishes, Marine Environmental Research, Volume 181, **2022**, 105732, ISSN 0141-1136.

---

## Author Response (AR2)

The paper has now incorporated the main points of the review and is conveying a clearer and stronger message. There are, however, two small points that I think could contribute to strengthen the paper:

1. I would suggest that the authors discuss how the fact that there are transients and climatic trends in the considered variables lead, together, to the observed variability. Because of that, I think that a paragraph discussing how to explore uncertainty in that variability, even mentioning as the authors said, that it will be part of a follow up study, will enrich the paper. I would suggest stressing the importance of exploring the possibility to estimate the ratio of climate signal to internal system variability and how the described modelling tool can contribute to that.

*Authors' response:* Thank you for your insightful comment. We appreciate your suggestion to discuss the role of transients and climatic trends in the observed variability. It is indeed important to explore uncertainty in this variability and estimate the ratio of climate signal to internal system variability. Our developed modelling framework allows us to quantify the proportion of climate signal to internal system variability. Transients and climatic trends within the system contribute to the observed variability through various processes. For example, changes in salinity in the lagoon can result from increased river inflow, precipitation, and periodic salt-water intrusions from the Baltic. To identify and quantify this variability, we will disassemble climatic projections and analyse model responses at different scales. Our upcoming follow-up study will employ statistical techniques such as the Mann-Kendall test, wavelet analysis, and time series decomposition. We will determine the most significant changes predicted by different RCM and RCP scenarios. By differentiating short-term fluctuations from long-term trends, we can evaluate the driving factors behind the observed variability and suggest appropriate mitigation measures.

2. The fact that the hydrodynamic model and the hydrological model (SHIFEM and SWAT) do not overlap may introduce a further uncertainty in the simulations. I think that the paper should discuss how those uncertainties could be reduced by an overlapping domain, some compatibility conditions or even bulk estimates of key control variables. That is becoming nowadays a very interesting topic where the proposed modelling tools could also contribute.

*Authors' response:* Thank you for raising the concern about model overlap. SWAT, as a hydrologic 1D model, is not deemed to be suitable for accurately representing the processes that are happening in a delta region. SHYFEM, being a hydrodynamic 3D model is better suited to represent such area. Hence, SWAT is used to represent its domain – basin-scale processes, while SHYFEM – shallow water/coastal processes. It would be complicated to check the overlapping area and confirm that the models are indeed "in sync", just because of the scale and the process representation. One such comparison could be the flow, as modelled by SHYFEM – flux of water thru the channel cross-section, and SWAT – outflow values for the same reach. Although this comparison would not entirely represent the same process, it can be performed to quantify the uncertainty in the representation of the overlapping domain. In the scope of this project – we did not assess such uncertainty, as we "cut" the model boundaries in the same place, meaning that there is no overlap of the domains, i.e., SHYFEM and SWAT do not overlap spatially because the output of SWAT (water discharge and temperature) serves as river boundary data for SHYFEM. This integration allows for the transfer of information between the models, ensuring that the hydrodynamic model captures relevant hydrological processes and responds to boundary conditions from SWAT.

*Authors' changes in manuscript:* We have created a separate section (4.9. Study limitations and uncertainty) within the Discussion section of the manuscript to specifically address the comments you raised. This approach allows for a focused discussion and helps ensure that the points you highlighted receive the attention they deserve. Additionally, we have carefully re-evaluated the content of our conclusions and decided to relocate certain parts of the conclusions to this newly created section. By doing so, we aim to

provide a comprehensive and cohesive response to your remarks while maintaining the logical flow of the paper. Incorporating your suggestions and implementing adjustments, we are confident that the clarity and effectiveness of our manuscript have been significantly improved.